# ON CONVERGENCE OF FEDERATED AVERAGING LANGEVIN DYNAMICS

## ABSTRACT

We propose a federated averaging Langevin algorithm (FA-LD) for uncertainty quantification and mean predictions with distributed clients. In particular, we generalize beyond normal posterior distributions and consider a general class of models. We develop theoretical guarantees for FA-LD for strongly log-concave distributions with non-i.i.d data and study how the injected noise and the stochastic-gradient noise, the heterogeneity of data, and the varying learning rates affect the convergence. Such an analysis sheds light on the optimal choice of local updates to minimize communication cost. Important to our approach is that the communication efficiency does not deteriorate with the injected noise in the Langevin algorithms. In addition, we examine in our FA-LD algorithm both independent and correlated noise used over different clients. We observe that there is also a trade-off between federation and communication cost there. As local devices may become inactive in the federated network, we also show convergence results based on different averaging schemes where only partial device updates are available.

## 1 INTRODUCTION

Federated learning (FL) allows multiple parties to jointly train a consensus model without sharing user data. Compared to the classical centralized learning regime, federated learning keeps training data on local clients, such as mobile devices or hospitals, where data privacy, security, and access rights are a matter of vital interest. This aggregation of various data resources heeding privacy concerns yields promising potential in areas of internet of things Chen et al. (2020), healthcare Li et al. (2020d; 2019b), text data Huang et al. (2020), and fraud detection Zheng et al. (2020).

A standard formulation of federated learning is a distributed optimization framework that tackles communication costs, client robustness, and data heterogeneity across different clients Li et al. (2020a). Central to the formulation is the efficiency of the communication, which directly motivates the communication-efficient federated averaging (FedAvg) McMahan et al. (2017). FedAvg introduces a global model to synchronously aggregate multi-step local updates on the available clients and yields distinctive properties in communication. However, FedAvg often stagnates at inferior local modes empirically due to the data heterogeneity across the different clients Charles & Konečný (2020); Woodworth et al. (2020). To tackle this issue, Karimireddy et al. (2020); Pathaky & Wainright (2020) proposed stateful clients to avoid the unstable convergence, which are, however, not scalable with respect to the number of clients in applications with mobile devices Al-Shedivat et al. (2021). In addition, the optimization framework often fails to quantify the uncertainty accurately for the parameters of interest, which are crucial for building estimators, hypothesis tests, and credible intervals. Such a problem leads to unreliable statistical inference and casts doubts on the credibility of the prediction tasks or diagnoses in medical applications.

To unify optimization and uncertainty quantification in federated learning, we resort to a *Bayesian treatment by sampling from a global posterior distribution*, where the latter is aggregated by infrequent communications from local posterior distributions. We adopt a popular approach for inferring posterior distributions for large datasets, the stochastic gradient Markov chain Monte Carlo (SG-MCMC) method Welling & Teh (2011); Vollmer et al. (2016); Teh et al. (2016); Chen et al. (2014); Ma et al. (2015), which enjoys theoretical guarantees beyond convex scenarios Raginsky et al. (2017); Zhang et al. (2017); Mangoubi & Vishnoi (2018); Ma et al. (2019). In particular, we examine in the federated learning setting the efficacy of the stochastic gradient Langevin dynamics (SGLD)

algorithm, which differs from stochastic gradient descent (SGD) in an additionally injected noise for exploring the posterior. The close resemblance naturally inspires us to adapt the optimization-based FedAvg to a distributed sampling framework. Similar ideas have been proposed in federated posterior averaging Al-Shedivat et al. (2021), where empirical study and analyses on Gaussian posteriors have shown promising potential of this approach. Compared to the appealing theoretical guarantees of optimization-based algorithms in federated learning Pathaky & Wainwright (2020); Al-Shedivat et al. (2021), the convergence properties of approximate sampling algorithms in federated learning is far less understood. To fill this gap, we proceed by asking the following question:

*Can we build a unified algorithm with convergence guarantees for sampling in FL?*

In this paper, we make a first step in answering this question in the affirmative. We propose the federated averaging Langevin dynamics (FA-LD) for posterior inference beyond the Gaussian distribution. We list our contributions as follows:

- We present a novel non-asymptotic convergence analysis for FA-LD from simulating strongly log-concave distributions on non-i.i.d data when the learning rate is fixed. The frequently used bounded gradient assumption of $\ell_2$ norm in FedAvg optimization is not required.
- The convergence analysis indicates that the injected noise, the data heterogeneity, and the stochastic-gradient noise are all driving factors that affect the convergence. Such an analysis also provides a concrete guidance on the optimal selection of the number of local updates.
- We present a convergence result for FA-LD with decaying learning rates. This strategy accelerates the computation by a logarithmic factor to achieve the precision $\epsilon$.
- The algorithm yields appealing extensions: (1) we can choose to inject either independent or correlated noise across local clients, yielding a trade-off between accuracy and efficacy of federation; (2) we can choose whether to activate all the devices to avoid the straggler's effect in real-world applications.

**Roadmap.** In Section 2, we discuss the related work and literature. In Section 3, we present the preliminary knowledge. In Section 4, we propose the federated averaging Langevin dynamics algorithm for posterior inference. In Section 5, we lay out the required assumptions, sketch the proof, and show the theoretical convergence results. In Section 6, we conclude our work.

## 2    RELATED WORK

**Federated Learning**    Current federated learning follows two paradigms. The first paradigm asks every client to learn the model using private data and communicate in model parameters. The second one uses encryption techniques to guarantee secure communication between clients. In this paper, we focus on the first paradigms Dean et al. (2012); Shokri & Shmatikov (2015); McMahan et al. (2016; 2017); Huang et al. (2021). There is a long list of works showing provable convergence algorithm for FedAvg types of algorithms in the field of optimization Li et al. (2020c; 2021); Huang et al. (2021); Khaled et al. (2019); Yu et al. (2019); Wang et al. (2019); Karimireddy et al. (2020). One line of research Li et al. (2020c); Khaled et al. (2019); Yu et al. (2019); Wang et al. (2019); Karimireddy et al. (2020) focuses on standard assumptions in optimization (such as, convex, smooth, strongly-convex, bounded gradient). The other line of work Li et al. (2021); Huang et al. (2021) proves the convergence in the regime where the model of interest is an over-parameterized neural network (also called NTK regime Jacot et al. (2018)).

**Scalable Monte Carlo methods**    SGLD Welling & Teh (2011) is the first stochastic gradient Monte Carlo method that tackles the scalability issue in big data problems. Ever since, variants of stochastic gradient Monte Carlo methods were proposed to accelerate the simulations by utilizing more general Markov dynamics Ma et al. (2015; 2018); Chen et al. (2014), Hessian approximation Ahn et al. (2012), parallel tempering Deng et al. (2020), as well as higher-order numerical schemes Chen et al. (2015); Li et al. (2019c); Cheng et al. (2018); Ma et al. (2021); Mou et al. (2021); Shen & Lee (2019).

**Distributed Monte Carlo methods**    Sub-posterior aggregation was initially proposed in Neiswanger et al. (2013); Wang & Dunson; Minsker et al. (2014) to accelerate MCMC methods to cope with large datasets. Other parallel MCMC algorithms Nishihara et al. (2014); Ahn et al. (2014);

Chen et al. (2016); Chowdhury & Jermaine (2018); Li et al. (2019a) propose to improve the efficiency of Monte Carlo computation in distributed or asynchronous systems. Gürbüzbalaban et al. (2021) proposed stochastic gradient Monte Carlo methods in decentralized systems. Al-Shedivat et al. (2021) introduced empirical studies of posterior averaging in federated learning.

**Notation**   For any positive integer $n$, we use $[n]$ to denote the set $\{1, 2, \cdots, n\}$. Let $N$ denote the number of clients. For each $c \in [N]$, we use $f^c$ and $\nabla f^c$ as the loss function and gradient of the function $f^c$ in client $c$. $\nabla \widetilde{f}^c(\cdot)$ is the *unbiased* stochastic gradient of $\nabla f^c$. In addition, we denote $p_c$ as the weight of the $c$-th client such that $p_c = \frac{n_c}{\sum_{i=1}^N n_i} \in (0, 1)$, where $n_c > 0$ is the number of data points in the $c$-th client. Let $T_\epsilon$ denote the number of global steps to achieve the precision $\epsilon$. Let $K$ denote the number of local steps and hence $T_\epsilon/K$ denotes the number of communications.

## 3   PRELIMINARIES

### 3.1   AN OPTIMIZATION PERSPECTIVE ON FEDERATED AVERAGING

Federated averaging (FedAvg) is a standard algorithm in federated learning and is typically formulated into a distributed optimization framework as follows

$$\min_\theta \ell(\theta) := \frac{\sum_{c=1}^N \ell^c(\theta)}{\sum_{c=1}^N n_c}, \quad \ell^c(\theta) := \sum_{i=1}^{n_c} l(\theta; x_{c,i}), \tag{1}$$

where $\theta \in \mathbb{R}^d$, $l(\theta; x_{c,j})$ is a certain loss function based on $\theta$ and the data point $x_{c,j}$.

One iterate of the FedAvg algorithm requires the following three steps:

- *Broadcast*: The center server *broadcasts* the latest model, $\theta_k$, to all local clients.
- *Local updates*: For any $c \in [N]$, the $c$-the client first sets $\theta_k^c = \theta_k$ and then conducts $K \geq 1$ local steps:

$$\beta_{k+1}^c = \theta_k^c - \eta \nabla \widetilde{\ell}^c(\theta_k^c),$$

  where $\eta$ is the learning rate and $\nabla \widetilde{\ell}^c$ is the unbiased estimate of the exact gradient $\nabla \ell^c$.
- *Synchronization*: The local models are aggregated into a unique model $\theta_{k+K} := \sum_{c=1}^N p_c \beta_{k+K}^c$ and sent to the center server.

From the optimization perspective, Li et al. (2020c) proved the convergence of the FedAvg algorithm on non-i.i.d data such that a larger number of local steps $K$ and a higher order of data heterogeneity slows down the convergence. Notably, Eq. (1) can be interpreted as maximizing the likelihood function, which is a special case of maximum a posteriori estimation (MAP) given a uniform prior.

### 3.2   STOCHASTIC GRADIENT LANGEVIN DYNAMICS

Posterior inference offers the exact uncertainty quantification ability of the predictions. A popular method for posterior inference with large dataset is the stochastic gradient Langevin dynamics (SGLD) Welling & Teh (2011), which injects additional noise into the stochastic gradient and adapts an optimization algorithm to a sampling one

$$\theta_{k+1} = \theta_k - \eta \nabla \widetilde{f}(\theta_k) + \sqrt{2\tau\eta}\xi_k,$$

where $\tau$ is the temperature and $\xi_k$ is a standard $d$-dimensional Gaussian vector. $f(\theta) := \sum_{c=1}^N \ell^c(\theta)$ is a energy function. $\widetilde{f}(\theta)$ is a unbiased estimate of $f(\theta)$. In the longtime limit, a well known result is that $\theta_k$ converges weakly to the distribution $\pi(\theta) \propto \exp(-f(\theta)/\tau)$ Teh et al. (2016) as $\eta \to 0$.

## 4   POSTERIOR INFERENCE VIA FEDERATED AVERAGING LANGEVIN DYNAMICS

The increasing concern for uncertainty estimation in federated learning motivates us to consider the simulation of the distribution $\pi(\theta) \propto \exp(-f(\theta)/\tau)$ with distributed clients.

**Problem formulation** We propose the federated averaging Langevin dynamics (FA-LD) based on the FedAvg framework in section 3.1. We follow the same *broadcast* step and *synchronization* step but propose to inject random noises for *local updates*. In particular, we consider the following scheme: for any $c \in [N]$, the $c$-the client first sets $\theta_k^c = \theta_k$ and then conducts $K \geq 1$ local steps:

$$\beta_{k+1}^c = \theta_k^c - \eta \nabla \widetilde{f}^c(\theta_k^c) + \sqrt{2\eta\tau}\Xi_k^c, \tag{2}$$

where $\nabla f^c(\theta) = \frac{1}{p_c}\nabla \ell^c(\theta)$. $\nabla \widetilde{f}^c(\theta)$ is the unbiased estimate of $\nabla f^c(\theta)$ and $\Xi_k^c$ is an independent Gaussian vector to be defined later.

Summing Eq. (2) from clients $c = 1$ to $N$, we have the aggregated stochastic process as follows

$$\beta_{k+1} = \theta_k - \eta \nabla \widetilde{f}(\theta_k) + \sqrt{2\eta\tau}\xi_k,$$

where

$$\beta_k = \sum_{c=1}^{N} p_c \beta_k^c, \quad \theta_k = \sum_{c=1}^{N} p_c \theta_k^c, \quad \nabla \widetilde{f}(\theta_k) = \sum_{c=1}^{N} p_c \nabla \widetilde{f}^c(\theta_k^c), \quad \xi_k = \sum_{c=1}^{N} p_c \Xi_k^c. \tag{3}$$

By the nature of the *synchronization* step, we always have $\beta_k = \theta_k$ whether $k + 1 \bmod K = 0$ or not. In what follows, we can write

$$\theta_{k+1} = \theta_k - \eta \nabla \widetilde{f}(\theta_k) + \sqrt{2\eta\tau}\xi_k, \tag{4}$$

which resembles the SGLD algorithm except that the construction of stochastic gradients is different and $\theta_k$ is *not accessible when $k \bmod K \neq 0$*. Since our target is to simulate from $\pi(\theta) \propto \exp(-f(\theta)/\tau)$, we expect that $\xi_k$ is a standard Gaussian vector. By the concentration property of independent Gaussian variables, it is natural to set $\Xi_k^c = \xi_k^c/\sqrt{p_c}$ so that $\xi_k = \sum_{c=1}^{N} p_c \Xi_k^c = \sum_{c=1}^{N} \sqrt{p_c}\xi_k^c$ and $\xi_k^c$ is also a standard Gaussian vector. Now we present it in Algorithm 1.

---

**Algorithm 1** Federated averaging Langevin dynamics algorithm (FA-LD), informal version of Algorithm 4. $\eta_k$ is the learning rate at iteration $k$. $\tau$ is the temperature. Denote by $\theta_k^c$ the model parameter in the $c$-th client at the $k$-th step. Denote the immediate result of one step SGLD update from $\theta_k^c$ by $\beta_k^c$. $\xi_k^c$ is an independent standard $d$-dimensional Gaussian vector at iteration $k$ for each client $c \in [N]$. A global synchronization is conducted every $K$ steps.

1:
$$\beta_{k+1}^c = \theta_k^c - \eta_k \nabla \widetilde{f}^c(\theta_k^c) + \sqrt{2\eta_k\tau/p_c}\xi_k^c, \tag{5}$$

2:
$$\theta_{k+1}^c = \begin{cases} \beta_{k+1}^c & \text{if } k + 1 \bmod K \neq 0 \\ \sum_{c=1}^{N} p_c \beta_{k+1}^c & \text{if } k + 1 \bmod K = 0. \end{cases} \tag{6}$$

---

**Algorithm 2** Hybrid federated averaging Langevin dynamics algorithm (hFA-LD), informal version of Algorithm 5. $\dot{\xi}_k$ is a $d$-dimensional Gaussian vector shared by all the clients; $\xi_k^c$ is an independent standard $d$-dimensional Gaussian vector at iteration $k$ for each client $c \in [N]$. $\rho$ denotes the correlation coefficient.

1:
$$\beta_{k+1}^c = \theta_k^c - \eta \nabla \widetilde{f}^c(\theta_k^c) + \sqrt{2\eta\tau\rho^2}\dot{\xi}_k + \sqrt{2\eta(1-\rho^2)\tau/p_c}\xi_k^c,$$

2:
$$\theta_{k+1}^c = \begin{cases} \beta_{k+1}^c & \text{if } k + 1 \bmod K \neq 0 \\ \sum_{c=1}^{N} p_c \beta_{k+1}^c & \text{if } k + 1 \bmod K = 0. \end{cases}$$

---

We observe that the local process in Eq. (5) maintains a temperature $\tau/p_c > \tau$ to converge to the stationary distribution $\pi$. Such a mechanism may limit the disclosure of individual data and shows a potential to ensure a higher level of privacy.

## 5 CONVERGENCE ANALYSIS

In this section, we show that FA-LD converges to the stationary distribution $\pi(\theta)$ in the 2-Wasserstein $(W_2)$ distance at a rate of $O(1/\sqrt{T_\epsilon})$ for strongly log-concave and smooth density. The $W_2$ distance is defined between a pair of Borel probability measures $\mu$ and $\nu$ on $\mathbb{R}^d$ as follows

$$W_2(\mu, \nu) := \inf_{\gamma^2 \in \text{Couplings}(\mu,\nu)} \left( \int \|\boldsymbol{\beta}_\mu - \boldsymbol{\beta}_\nu\|_2^2 d\gamma^2(\boldsymbol{\beta}_\mu, \boldsymbol{\beta}_\nu) \right)^{\frac{1}{2}},$$

where $\| \cdot \|_2$ denotes the $\ell_2$ norm on $\mathbb{R}^d$ and the pair of random variables $(\boldsymbol{\beta}_\mu, \boldsymbol{\beta}_\nu) \in \mathbb{R}^d \times \mathbb{R}^d$ is a coupling with the marginals following $\mathcal{L}(\boldsymbol{\beta}_\mu) = \mu$ and $\mathcal{L}(\boldsymbol{\beta}_\nu) = \nu$. Note that $\mathcal{L}(\cdot)$ denotes a distribution of a random variable. Such a distance is more appealing than the total variation or the Kullback–Leibler divergence in statistical machine learning applications for providing the estimates of the first and second order moments.

### 5.1 NOTATION AND ASSUMPTIONS

We make standard assumptions on the smoothness and convexity of the functions $f^1, f^2, \cdots, f^N$, which naturally yields appealing tail properties of the stationary measure $\pi$. Thus, we no longer require a restrictive assumption on the bounded gradient in $\ell_2$ norm as in Koloskova et al. (2019); Yu et al. (2019); Li et al. (2020c). In addition, to control the distance between $\nabla f^c$ and $\nabla \widetilde{f}^c$, we also assume a bounded variance of the stochastic gradient in assumption 5.3.

**Assumption 5.1** (Smoothness). *For each $c \in [N]$, we say $f^c$ is L-smooth if for some $L > 0$*

$$f^c(y) \leq f^c(x) + \langle \nabla f^c(x), y - x \rangle + \frac{L}{2}\|y - x\|_2^2 \quad \forall x, y \in \mathbb{R}^d.$$

**Assumption 5.2** (Strongly convex). *For each $c \in [N]$, $f^c$ is m-strongly convex if for some $m > 0$*

$$f^c(x) \geq f^c(y) + \langle \nabla f^c(y), x - y \rangle + \frac{m}{2}\|y - x\|_2^2 \quad \forall x, y \in \mathbb{R}^d.$$

**Assumption 5.3** (Bounded variance, informal version of Assumption A.3). *For each $c \in [N]$, the variance of noise in the stochastic gradient $\nabla \widetilde{f}^c(x)$ in each client is upper bounded such that*

$$\mathbb{E}[\|\nabla \widetilde{f}^c(x) - \nabla f^c(x)\|_2^2] \leq \sigma^2 d, \quad \forall x \in \mathbb{R}^d.$$

**Quality of non-i.i.d data** Denote by $\theta_*$ the global minimum of $f$. Next, we quantify the degree of the non-i.i.d data by $\gamma := \max_{c \in [N]} \|\nabla f^c(\theta_*)\|_2$, which is a non-negative constant and yields a larger scale if the data is less identically distributed.

### 5.2 PROOF SKETCH

The proof hinges on showing the one-step result in the $W_2$ distance. To facilitate the analysis, we first define an auxiliary continuous-time processes $(\bar{\theta}_t)_{t \geq 0}$ without communication concerns

$$d\bar{\theta}_t = -\nabla f(\bar{\theta}_t)dt + \sqrt{2\tau}d\overline{W}_t, \tag{7}$$

where $\bar{\theta}_t = \sum_{c=1}^N p_c \bar{\theta}_t^c$, $\nabla f(\bar{\theta}_t) = \sum_{c=1}^N p_c \nabla f^c(\bar{\theta}_t^c)$, $\bar{\theta}_t^c$ is the continuous-time variable at client $c$, and $\overline{W}$ is a $d$-dimensional Brownian motion. The continuous-time algorithm is known to converge to the stationary distribution $\pi(\theta) \propto e^{-\frac{f(\theta)}{\tau}}$, where $f(\theta) = \sum_{c=1}^N p_c f^c(\theta)$. Assume that $\bar{\theta}_0$ simulates from the stationary distribution $\pi$, then it follows that $\bar{\theta}_t \sim \pi$ for any $t \geq 0$.

#### 5.2.1 DOMINATED CONTRACTION IN FEDERATED LEARNING

The first target is to show a certain contraction property of $\|\beta - \theta - \eta(\nabla f(\beta) - \nabla f(\theta))\|_2^2$ based on distributed clients with infrequent communications. Consider a standard decomposition

$$\|\beta - \theta - \eta(\nabla f(\beta) - \nabla f(\theta))\|_2^2$$

$$= \|\beta - \theta\|_2^2 - 2\eta \underbrace{\langle \beta - \theta, \nabla f(\beta) - \nabla f(\theta)\rangle}_{\mathcal{I}} + \eta^2 \|\nabla f(\beta) - \nabla f(\theta)\|_2^2.$$

Using Eq.(3), we decompose $\mathcal{I}$ and apply Jensen's inequality to obtain the lower bound of $\mathcal{I}$. In what follows, we have the following lemma.

**Lemma 5.4** (Dominated contraction property, informal version of Lemma B.1). *Assume assumptions 5.1 and 5.2 hold. For any learning rate $\eta \in (0, \frac{1}{L+m}]$, any $\{\theta^c\}_{c=1}^N, \{\beta^c\}_{c=1}^N \in \mathbb{R}^d$, we have*

$$\|\beta - \theta - \eta(\nabla f(\beta) - \nabla f(\theta))\|_2^2 \leq (1 - \eta m) \cdot \|\beta - \theta\|_2^2 + 4\eta L \sum_{c=1}^N p_c \cdot \underbrace{(\|\beta^c - \beta\|_2^2 + \|\theta^c - \theta\|_2^2)}_{\text{divergence term}},$$

*where $\beta = \sum_{c=1}^N p_c \beta^c$, $\theta = \sum_{c=1}^N p_c \theta^c$, $\nabla f(\theta) = \sum_{c=1}^N p_c \nabla f^c(\theta^c)$, and $\nabla f(\beta) = \sum_{c=1}^N p_c \nabla f^c(\beta^c)$. It implies that as long as the local parameters $\theta^c, \beta^c$ and global $\theta, \beta$ don't differ each other too much, we can guarantee the desired convergence. In a special case when the communication is conducted at every iteration, the divergence term disappears and recovers the standard contraction* Dalalyan & Karagulyan (2019).

### 5.2.2 BOUNDING DIVERGENCE

The following result shows that given a finite number of local steps $K$, the divergence between $\theta^c$ in local client and $\theta$ in the center is bounded in $\ell_2$ norm. Notably, since the Brownian motion leads to a lower order term $O(\eta)$ instead of $O(\eta^2)$, a naïve proof framework such as Li et al. (2020c) may lead to a crude upper bound for the final convergence.

**Lemma 5.5** (Bounded divergence, informal version of Lemma B.3). *Assume assumptions 5.1, 5.2, and 5.3 hold. For any learning rate $\eta \in (0, 2/m)$ and $\|\theta_0^c - \theta_*\|_2^2 \leq d\mathcal{D}^2$ for any $c \in [N]$, we have the $\ell_2$ upper bound of the divergence between local clients and the center as follows*

$$\sum_{c=1}^N p_c \mathbb{E}\|\theta_k^c - \theta_k\|_2^2 \leq O(K^2\eta^2 d) + O(K\eta d).$$

The result also relies on showing a uniform upper bound in $\ell_2$ norm, which avoids making extra bounded gradient assumptions.

### 5.2.3 COUPLING TO THE STATIONARY PROCESS

Note that $\bar{\theta}_t$ is initialized from the stationary distribution $\pi$. The solution to the continuous-time process Eq.(7) follows:

$$\bar{\theta}_t = \bar{\theta}_0 - \int_0^t \nabla f(\bar{\theta}_s)\mathrm{d}s + \sqrt{2\tau} \cdot \overline{W}_t, \qquad \forall t \geq 0. \tag{8}$$

Set $t \to (k+1)\eta$ and $\bar{\theta}_0 \to \bar{\theta}_{k\eta}$ for Eq.(8) and consider a *synchronous coupling* such that $W_{(k+1)\eta} - W_{k\eta} := \sqrt{\eta}\xi_k$ is used to cancel the noise terms, we have

$$\bar{\theta}_{(k+1)\eta} = \bar{\theta}_{k\eta} - \int_{k\eta}^{(k+1)\eta} \nabla f(\bar{\theta}_s)\mathrm{d}s + \sqrt{2\tau\eta}\xi_k. \tag{9}$$

Subtracting Eq.(4) from Eq.(9) and taking square and expectation on both sides yield that

$$\mathbb{E}\|\bar{\theta}_{(k+1)\eta} - \theta_{k+1}\|_2^2 \leq (1 - \eta m/2) \cdot \mathbb{E}\|\bar{\theta}_{k\eta} - \theta_k\|_2^2 + \text{divergence term} + \text{time error}.$$

Eventually, we arrive at the one-step error bound for establishing the convergence results.

**Lemma 5.6** (One step update, informal version of Lemma B.5). *Assume assumptions 5.1, 5.2, and 5.3 hold. Consider Algorithm 1 with any learning rate $\eta \in (0, \frac{1}{2L})$ and $\|\theta_0^c - \theta_*\|_2^2 \leq d\mathcal{D}^2$ for any $c \in [N]$, where $\theta_*$ is the global minimum for the function $f$. Then*

$$W_2^2(\mu_{k+1}, \pi) \leq (1 - \eta m/2) \cdot W_2^2(\mu_k, \pi) + O(\eta^2 d(K^2 + \kappa)),$$

*where $\mu_k$ denotes the probability measure of $\theta_k$ and $\kappa = L/m$ is the condition number.*

## 5.3 FULL DEVICE PARTICIPATION

### 5.3.1 CONVERGENCE BASED ON INDEPENDENT NOISE

When the synchronization step is conducted at every iteration $k$, the FA-LD algorithm is essentially the standard SGLD algorithm Welling & Teh (2011). Theoretical analysis based on the 2-Wasserstein distance has been established in Durmus & Moulines (2016); Dalalyan (2017a); Dalalyan & Karagulyan (2019). However, in scenarios of $K > 1$ with distributed clients, a divergence between the global variable $\theta_k$ and local variable $\theta_k^c$ appears and unavoidably affects the performance. The upper bound on the sampling error is presented as follows.

**Theorem 5.7** (Main result, informal version of Theorem B.6). *Assume assumptions 5.1, 5.2, and 5.3 hold. Consider Algorithm 1 with a fixed learning rate $\eta \in (0, \frac{1}{2L}]$ and $\|\theta_0^c - \theta_*\|_2^2 \leq d\mathcal{D}^2$ for any $c \in [N]$, we have* [†]

$$W_2(\mu_k, \pi) \leq (1 - \eta m/4)^k \cdot \left( \sqrt{2d}(\mathcal{D} + \sqrt{\tau/m}) \right) + 30\kappa\sqrt{\eta md} \cdot \sqrt{(K^2 + \kappa)H_0}.$$

*where $\mu_k$ denotes the probability measure of $\theta_k$ at iteration $k$, $K$ denotes the number of local updates, $\kappa := L/m$, $\gamma := \max_{c \in [N]} \|\nabla f^c(\theta_*)\|_2$, and $H_0 := \mathcal{D}^2 + \max_{c \in [N]} \frac{\tau}{mp_c} + \frac{\gamma^2}{m^2 d} + \frac{\sigma^2}{m^2}$.*

We observe that the initialization, the scale of the injected noise, the heterogeneity of the data, and the noise in the stochastic gradient all affect the convergence. Similar to the result of Li et al. (2020c), FA-LD with $K$-local steps resembles the behaviour of one-step SGLD with a large learning rate.

**Optimal choice of $K$.** To ensure the algorithm to achieve the precision $\epsilon$ based on the total number of steps $T_\epsilon$ and the learning rate $\eta$, we can set

$$30\kappa\sqrt{\eta md} \cdot \sqrt{(K^2 + \kappa)H_0} \leq \epsilon/2, \quad \exp\left(-\frac{\eta m}{4}T_\epsilon\right) \cdot \sqrt{2d}(\mathcal{D} + \sqrt{\tau/m}) \leq \epsilon/2.$$

This readily leads to

$$\eta m \leq O\left(\frac{\epsilon^2}{d\kappa^2(K^2 + \kappa)H_0}\right), \quad T_\epsilon \geq \Omega\left(\frac{\log(d/\epsilon)}{m\eta}\right).$$

Plugging into the upper bound of $\eta m$, it implies that to reach the precision $\epsilon$, it suffices to set

$$T_\epsilon = \Omega(\epsilon^{-2} d\kappa^2 (K^2 + \kappa)H_0 \cdot \log(d/\epsilon)). \tag{10}$$

It's obvious that $H_0 = \Omega(\mathcal{D}^2) = \Omega(1)$, thus we can conclude that the number of communication rounds is around the order

$$\frac{T_\epsilon}{K} = \Omega\left(K + \frac{\kappa}{K}\right),$$

where the value of $\frac{T_\epsilon}{K}$ first decreases and then increases with respect to $K$, indicating that setting $K$ either too large or too small may lead to high communication costs and hurt the performance. Ideally, $K$ should be selected in the scale of $\Omega(\sqrt{\kappa})$. Combining the definition of $T_\epsilon$ in Eq. (10), this suggests an interesting result that the optimal $K$ for FA-LD should be in the order of $O(\sqrt{T_\epsilon})$. Similar results have been achieved by Stich (2019); Li et al. (2020c).

### 5.3.2 CONVERGENCE GUARANTEES VIA VARYING LEARNING RATES

**Theorem 5.8** (Informal version of Theorem B.7). *Assume assumptions 5.1, 5.2, and 5.3 hold. Consider Algorithm 1 with an initialization satisfying $\|\theta_0^c - \theta_*\|_2^2 \leq d\mathcal{D}^2$ for any $c \in [N]$ and the varying learning rate following*

$$\eta_k = \frac{1}{2L + (1/12)mk}, \quad k = 1, 2, \cdots.$$

*Then for any $k \geq 0$, we have*

$$W_2(\mu_k, \pi) \leq 45\kappa\sqrt{(K^2 + \kappa)H_0} \cdot (\eta_k md)^{1/2}, \quad \forall k \geq 0.$$

[†]For ease of presentation, we report the result based on $K^2$ instead of $(K-1)^2$. The upper bound based on $(K-1)^2$ is detailed in the supplementary file.

Note that the above result implies that to achieve the precision $\epsilon$, we require

$$W_2(\mu_k, \pi) \le 45\kappa\sqrt{(K^2 + \kappa)H_0} \cdot \left(\frac{md}{2L + (1/12)mk}\right)^{1/2} \le \epsilon.$$

We therefore require $\Omega(\epsilon^{-2}d)$ iterations to achieve the precision $\epsilon$, which improves the $\Omega(\epsilon^{-2}d\log(d/\epsilon))$ rate for FA-LD with a fixed learning rate by a $O(\log(d/\epsilon))$ factor.

### 5.3.3 PRIVACY-ACCURACY TRADE-OFF VIA CORRELATED NOISES

Note that Algorithm 1 requires all the local clients to generate the independent noise $\xi_k^c$. Such a mechanism enjoys the convenience of the implementation and yields a potential to protect the privacy of data and alleviates the security issue. However, the large scale noise inevitable slows down the convergence. To handle this issue, the independent noise can be generalized to correlated noise based on a correlation coefficient $\rho$ between different clients. Replacing Eq. (5) with

$$\beta_{k+1}^c = \theta_k^c - \eta\nabla\widetilde{f}^c(\theta_k^c) + \sqrt{2\eta\tau\rho^2}\dot{\xi}_k + \sqrt{2\eta(1-\rho^2)\tau/p_c}\xi_k^c, \tag{11}$$

where $\dot{\xi}_k$ is a $d$-dimensional standard Gaussian vector shared by all the clients at iteration $k$ and $\dot{\xi}_k$ is dependent with $\xi_k^c$ for any $c \in [N]$. Following the synchronization step based on Eq. (6), we have

$$\theta_{k+1} = \theta_k - \eta\nabla\widetilde{f}(\theta_k) + \sqrt{2\eta\tau}\xi_k, \tag{12}$$

where $\xi_k = \rho\dot{\xi}_k + \sqrt{1-\rho^2}\sum_{c=1}^N \sqrt{p_c}\xi_k^c$. Since the variance of i.i.d variables is additive, it is clear that $\xi_k$ follows the standard $d$-dimensional Gaussian distribution. The inclusion of the correlated noise implicitly reduces the temperature for each client and naturally yields a trade-off between federation and accuracy. We refer to the algorithm with correlated noise as the hybrid federated averaging Langevin dynamics (hFA-LD) and present it in Algorithm 2.

Since the inclusion of correlated noise doesn't affect the iterate of Eq. (12), the algorithm property maintains the same except the scale of the temperature $\tau$ and efficacy of federation are changed. Based on a target correlation coefficient $\rho \ge 0$, Eq. (11) is equivalent to applying a temperature $T_{c,\rho} = \tau(\rho^2 + (1-\rho^2)/p_c)$. In particular, setting $\rho = 0$ leads to $T_{c,0} = \tau/p_c$, which exactly recovers Algorithm 1; however, setting $\rho = 1$ leads to $T_{c,1} = \tau$, where the injected noise in local clients is reduced by $1/p_c$ times. Now we adjust the analysis as follows

**Theorem 5.9** (Informal version of Theorem B.8). *Assume assumptions 5.1, 5.2, and 5.3 hold. Consider Algorithm 2 with a correlation coefficient $\rho \in [0, 1]$, $\eta \in (0, \frac{1}{2L}]$ and $\|\theta_0^c - \theta_*\|_2^2 \le d\mathcal{D}^2$ for any $c \in [N]$, we have*

$$W_2(\mu_k, \pi) \le (1 - \eta m/4)^k \cdot \left(\sqrt{2d}\big(\mathcal{D} + \sqrt{\tau/m}\big)\right) + 30\kappa\sqrt{\eta md} \cdot \sqrt{(K^2 + \kappa)H_\rho},$$

*where $\mu_k$ denotes the probability measure of $\theta_k$, $H_\rho := \mathcal{D}^2 + \frac{1}{m}\max_{c\in[N]} T_{c,\rho} + \frac{\gamma^2}{m^2d} + \frac{\sigma^2}{m^2}$.*

Such a mechanism leads to a trade-off between the efficacy of federation and accuracy and motivates us to exploit the optimal $\rho$ under the differential-privacy theories Wang et al. (2015).

## 5.4 PARTIAL DEVICE PARTICIPATION

Full device participation enjoys appealing convergence properties. However, it suffers from the straggler's effect in real-world applications, where the communication is limited by the slowest device. Partial device participation handles this issue by only allowing a small portion of devices in each communication and greatly increased the communication efficiency in a federated network.

The first device-sampling scheme I Li et al. (2020b) selects a total of $S$ devices, where the $c$-th device is selected with a probability $p_c$. The first theoretical justification for convex optimization has been proposed by Li et al. (2020c).

**(Scheme I: with replacement).** Assume $\mathcal{S}_k = \{n_1, n_2, \cdots, n_S\}$, where $n_j \in [N]$ is a random number that takes a value of $c$ with a probability $p_c$ for any $j \in \{1, 2, \cdots, S\}$. The synchronization step follows that $\theta_k = \frac{1}{S}\sum_{c\in\mathcal{S}_k}\theta_k^c$.

Another strategy is to uniformly select $S$ devices without replacement. We follow Li et al. (2020c) and assume $S$ indices are selected uniformly without replacement. In addition, the convergence also requires an additional assumption on balanced data Li et al. (2020c).

**(Scheme II: without replacement).** Assume $\mathcal{S}_k = \{n_1, n_2, \cdots, n_S\}$, where $n_j \in [N]$ is a random number that takes a value of $c$ with a probability $\frac{1}{S}$ for any $j \in \{1, 2, \cdots, S\}$. Assume the data is balanced such that $p_1 = \cdots = p_N = \frac{1}{N}$. The synchronization step follows that $\theta_k = \frac{N}{S} \sum_{c \in \mathcal{S}_k} p_c \theta_k^c = \frac{1}{S} \sum_{c \in \mathcal{S}_k} \theta_k^c$.

---

**Algorithm 3** Hybrid federated Averaging Langevin dynamics Algorithm (FA-LD) with partial device participation, informal version of Algorithm 6. $\mathcal{S}_k$ is sampled according to a device-sampling rule based on scheme I or II.

1:
$$\beta_{k+1}^c = \theta_k^c - \eta \nabla \widetilde{f}^c(\theta_k^c) + \sqrt{2\eta\tau\rho^2}\dot{\xi}_k + \sqrt{2\eta(1-\rho^2)\tau/p_c}\xi_k^c,$$

2:
$$\theta_{k+1}^c = \begin{cases} \beta_{k+1}^c & \text{if } k+1 \bmod K \neq 0 \\[2mm] \sum_{c \in \mathcal{S}_{k+1}} \frac{1}{S}\beta_{k+1}^c & \text{if } k+1 \bmod K = 0. \end{cases}$$

---

**Theorem 5.10** (Informal version of Theorem C.3). *Assume assumptions 5.1, 5.2, and 5.3 hold. Consider Algorithm 3 with a hyperparameter $\rho \in [0, 1]$, a fixed learning rate $\eta \in (0, \frac{1}{2L}]$ and $\|\theta_0^c - \theta_*\|_2^2 \leq d\mathcal{D}^2$ for any $c \in [N]$, we have*

$$W_2(\mu_k, \pi) \leq (1 - \eta m/4)^k \cdot \left( \sqrt{2d}\left(\mathcal{D} + \sqrt{\tau/m}\right) \right)$$

$$+ 30\kappa\sqrt{\eta md} \cdot \sqrt{H_\rho(K^2 + \kappa)} + O\left( \sqrt{\frac{d}{S}(\rho^2 + N(1-\rho^2))C_S} \right),$$

*where $C_S = 1$ for Scheme I and $C_S = \frac{N-S}{N-1}$ for Scheme II.*

We observe that partial device participation leads to an extra bias regardless of the scale of $\eta$. To reduce such a bias, we suggest to consider highly correlated injected noise, such as $\rho = 1$, to reduce the impact of the injected noise. By setting $O(\sqrt{d/S}) \leq \epsilon/3$ and following a similar learning rate as in section 5.3.1, we can achieve the precision $\epsilon$ within $\Omega(\epsilon^{-2}d\log(d/\epsilon))$ iterations given a large number of devices satisfying $S = \Omega(\epsilon^{-2}d)$.

The device-sampling scheme I provides a viable solution to handle the straggler's effect in full device participation and greatly accelerates the communication efficiency. In addition, scheme I is rather robust to the data heterogeneity and doesn't require the data to be balanced. In other words, this device-sampling scheme is more preferred if a system is free to activate any devices at any time.

In more practical cases where a system can only operate based on the first $S$ messages for the local updates. The device-sampling scheme II proposes a concrete treatment to tackle this issue. Given a balanced data across different clients and each device is uniformly sampled, we can achieve a reasonable approximation. If $S = 1$, our Scheme II matches the result in the Scheme I. If $S = N$, then our Scheme II recovers the result in the full device setting. If $S = N - o(N)$, then our Scheme II bound is better than scheme I.

## 6 CONCLUSION AND FUTURE WORK

We propose a novel convergence analysis for federated averaging Langevin dynamics (FA-LD) with distributed clients. Our results no longer require the bounded gradient assumption in $\ell_2$ norm as in the optimization-driven literature in federated learning. The theoretical guarantees yield a concrete guidance on the selection of the optimal number of local updates. In addition, the convergence highly depends on the data heterogeneity and the injected noises, where the latter also inspires us to consider correlated injected noise to balance between the efficacy of federation and accuracy.

Our work initiated the theoretical study of standard sampling algorithms in federated learning and paved the way for future works of advanced Monte Carlo methods, such as underdamped Langevin dynamics Cheng et al. (2018), replica exchange Monte Carlo (also known as parallel tempering) Deng et al. (2020) in federated learning. It is also interesting to study the optimal number of local steps under the non-strongly convex Dalalyan (2017b) or non-convex assumptions Raginsky et al. (2017); Ma et al. (2019).

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

**Roadmap.** In Section A, we layout the formulation of the algorithm, basic notations, and definitions. In Section B, we present the main convergence analysis for full device participation. We discuss the optimal number of local updates based on a fixed learning rate, the acceleration achieved by varying learning rates, and the privacy-accuracy trade-off through correlated noises. In Section C, we analyze the convergence of partial device participation through two device-sampling schemes. In Section D, we provide lemmas to upper bound the contraction, discretization and divergence for proving the main convergence results. In Section E, we include supporting lemmas to prove results in the previous section. In Section F, we establish the initial condition.

## A  PRELIMINARIES

### A.1  BASIC NOTATIONS AND BACKGROUNDS

Let $N$ denote the number of clients. Let $T_\epsilon$ denote the number of global steps to achieve the precision $\epsilon$. Let $K$ denote the number of local steps. For each $c \in [N] := \{1, 2, \cdots, N\}$, we use $f^c$ and $\nabla f^c$ denote the loss function and gradient of the function $f^c$ in client $c$. For the stochastic gradient oracle, we denote by $\nabla \widetilde{f}^c(\cdot)$ the *unbiased* estimate of the exact gradient $\nabla f^c$ of client $c$. In addition, we denote $p_c$ as the weight of the $c$-th client such that $p_c \geq 0$ and $\sum_{c=1}^N p_c = 1$. $\xi_k^c$ is an independent standard $d$-dimensional Gaussian vector at iteration $k$ for each client $c \in [N]$ and $\dot{\xi}_k$ is a unique Gaussian vector shared by all the clients.

---

**Algorithm 4** Federated averaging Langevin dynamics algorithm (FA-LD). Denote by $\theta_k^c$ the model parameter in the $c$-th client at the $k$-th step. Denote the immediate result of one step SGLD update from $\theta_k^c$ by $\beta_k^c$. $\xi_k^c$ is an independent standard $d$-dimensional Gaussian vector at iteration $k$ for each client $c \in [N]$. A global synchronization is conducted every $K$ steps. This is a complete version of Algorithm 1.

1:
$$\beta_{k+1}^c = \theta_k^c - \eta \nabla \widetilde{f}^c(\theta_k^c) + \sqrt{2\eta\tau/p_c}\xi_k^c, \tag{13}$$

2:
$$\theta_{k+1}^c = \begin{cases} \beta_{k+1}^c & \text{if } k+1 \bmod K \neq 0 \\[2mm] \sum_{c=1}^N p_c\beta_{k+1}^c & \text{if } k+1 \bmod K = 0. \end{cases} \tag{14}$$

---

Inspired by Li et al. (2020c), we define two virtual sequences

$$\beta_k = \sum_{c=1}^N p_c\beta_k^c, \qquad \theta_k = \sum_{c=1}^N p_c\theta_k^c, \tag{15}$$

which are *both inaccessible when k mod K $\neq 0$*. For the gradients and injected noise, we also define

$$\nabla f(\theta_k) = \sum_{c=1}^N p_c \nabla f^c(\theta_k^c), \quad \nabla\widetilde{f}(\theta_k) = \sum_{c=1}^N p_c \nabla\widetilde{f}^c(\theta_k^c), \quad \xi_k = \sum_{c=1}^N \sqrt{p_c}\xi_k^c. \tag{16}$$

In what follows, it is clear that $\mathbb{E}\nabla\widetilde{f}(\theta) = \sum_{c=1}^N p_c \mathbb{E}\nabla\widetilde{f}^c(\theta^c) = \nabla f(\theta)$ for any $\theta^c \in \mathbb{R}^d$ and any $c \in [N]$. Summing Eq.(13) from clients $c = 1$ to $N$ and combining Eq.(15) and Eq.(16), we have

$$\beta_{k+1} = \theta_k - \eta \nabla\widetilde{f}(\theta_k) + \sqrt{2\eta\tau}\xi_k. \tag{17}$$

Moreover, we always have $\beta_k = \theta_k$ whether $k+1 \bmod E = 0$ or not by Eq.(14) and Eq.(15). In what follows, we can write

$$\theta_{k+1} = \theta_k - \eta \nabla\widetilde{f}(\theta_k) + \sqrt{2\eta\tau}\xi_k, \tag{18}$$

which resembles the SGLD algorithm Welling & Teh (2011) except that the construction of stochastic gradients is different and $\theta_k$ is *not accessible when k mod K $\neq 0$*. To facilitate the analysis, we also define an auxiliary continuous-time processes $(\bar{\theta}_t)_{t\geq 0}$

$$\mathrm{d}\bar{\theta}_t = -\nabla f(\bar{\theta}_t) \cdot \mathrm{d}t + \sqrt{2\tau} \cdot \mathrm{d}\overline{W}_t, \tag{19}$$

where $\bar{\theta}_t = \sum_{c=1}^{N} p_c \bar{\theta}_t^c$, $\nabla f(\bar{\theta}_t) = \sum_{c=1}^{N} p_c \nabla f^c(\bar{\theta}_t^c)$, $\bar{\theta}_t^c$ is the continuous-time variable at client $c$, and $\overline{W}$ is a $d$-dimensional Brownian motion. The continuous-time algorithm is referred to as Federated Averaging Langevin diffusion and is described as

$$\mathrm{d}\bar{\beta}_t^c = -\nabla f^c(\bar{\theta}_t^c) \cdot \mathrm{d}t + \sqrt{2\tau/p_c} \cdot \mathrm{d}\overline{W}_t^c$$

$$\bar{\theta}_t^c = \sum_{c=1}^{N} p_c \bar{\beta}_t^c.$$

Since the synchronization step is conducted at every time step $t$, the Federated Averaging Langevin diffusion performs the same as the standard Langevin diffusion with the temperature $\tau$ and convergences to the stationary distribution $\pi(\theta) \propto \exp(-f(\theta)/\tau)$, where $f(\theta) = \sum_{c=1}^{N} p_c f^c(\theta)$. Assume that $\bar{\theta}_0$ simulates from the stationary distribution $\pi$, then it follows that $\bar{\theta}_t \sim \pi$ for any $t \geq 0$.

## A.2 ASSUMPTIONS AND DEFINITIONS

**Assumption A.1** (Smoothness). *For each $c \in [N]$, we say $f^c$ is $L$-smooth if for some $L > 0$*

$$f^c(y) \leq f^c(x) + \langle \nabla f^c(x), y - x \rangle + \frac{L}{2}\|y - x\|_2^2 \quad \forall x, y \in \mathbb{R}^d.$$

Note that the above assumption is equivalent to saying that

$$\|\nabla f^c(y) - \nabla f^c(x)\|_2 \leq L\|y - x\|_2, \quad \forall x, y \in \mathbb{R}^d.$$

**Assumption A.2** (Strong convexity). *For each $c \in [N]$, $f^c$ is $m$-strongly convex if for some $m > 0$*

$$f^c(x) \geq f^c(y) + \langle \nabla f^c(y), x - y \rangle + \frac{m}{2}\|y - x\|_2^2 \quad \forall x, y \in \mathbb{R}^d.$$

An alternative formulation for strong convexity is that

$$\langle \nabla f^c(x) - \nabla f^c(y), x - y \rangle \geq m \|x - y\|_2^2 \quad \forall x, y \in \mathbb{R}^d.$$

**Assumption A.3** (Bounded variance, restatement of Assumption 5.3). *For each $c \in [N]$, the variance of noise in the stochastic gradient $\nabla \widetilde{f}^c(x)$ in each client is upper bounded such that*

$$\mathbb{E}[\|\nabla \widetilde{f}^c(x) - \nabla f^c(x)\|_2^2] \leq \sigma^2 d, \quad \forall x \in \mathbb{R}^d.$$

The bounded variance in the stochastic gradient is a rather standard assumption and has been widely used in Cheng et al. (2018); Dalalyan & Karagulyan (2019); Li et al. (2020c). Extension of bounded variance to unbounded cases such as $\mathbb{E}[\|\nabla \widetilde{f}^c(x) - \nabla f^c(x)\|_2^2] \leq \delta(L^2 x^2 + B^2)$ for some $M$ and $\delta \in [0, 1)$ is quite straightforward and has been adopted in assumption A.4 stated in Raginsky et al. (2017). The proof framework remains the same.

**Quality of non-i.i.d data**    Denote by $\theta_*$ the global minimum of $f$. Next, we quantify the degree of the non-i.i.d data by $\gamma := \max_{c \in [N]} \|\nabla f^c(\theta_*)\|_2$, which is a non-negative constant and yields a smaller scale if the data is more evenly distributed.

**Definition A.4.** *We define parameter $T_{c,\rho}$ $H_\rho^2$, $\kappa$ and $\gamma^2$*

$$T_{c,\rho} := \tau(\rho^2 + (1 - \rho^2)/p_c),$$

$$H_\rho := \underbrace{\mathcal{D}^2}_{\text{initialization}} + \underbrace{\frac{1}{m}\max_{c \in [N]} T_{c,\rho}}_{\text{injected noise}} + \underbrace{\frac{\gamma^2}{m^2 d}}_{\text{data heterogeneity}} + \underbrace{\frac{\sigma^2}{m^2}}_{\text{stochastic noise}},$$

$$\kappa := L/m,$$

$$\gamma^2 := \max_{c \in [N]} \|\nabla f^c(\theta_*)\|_2^2.$$

## B FULL DEVICE PARTICIPATION

### B.1 ONE-STEP UPDATE

**Wasserstein distance** We define the 2-Wasserstein distance between a pair of Borel probability measures $\mu$ and $\nu$ on $\mathbb{R}^d$ as follows

$$W_2(\mu, \nu) := \inf_{\gamma^2 \in \text{Couplings}(\mu, \nu)} \left( \int \|\boldsymbol{\beta}_\mu - \boldsymbol{\beta}_\nu\|_2^2 d\gamma^2(\boldsymbol{\beta}_\mu, \boldsymbol{\beta}_\nu) \right)^{\frac{1}{2}},$$

where $\|\cdot\|_2$ denotes the $\ell_2$ norm on $\mathbb{R}^d$ and the pair of random variables $(\boldsymbol{\beta}_\mu, \boldsymbol{\beta}_\nu) \in \mathbb{R}^d \times \mathbb{R}^d$ is a coupling with the marginals following $\mathcal{L}(\boldsymbol{\beta}_\mu) = \mu$ and $\mathcal{L}(\boldsymbol{\beta}_\nu) = \nu$. $\mathcal{L}(\cdot)$ denotes a distribution of a random variable.

The following result provides a crucial contraction property based on distributed clients with infrequent synchronizations.

**Lemma B.1** (Dominated contraction property, restatement of Lemma 5.4). *Assume assumptions A.1 and A.2 hold. For any learning rate $\eta \in (0, \frac{1}{L+m}]$, any $\{\theta^c\}_{c=1}^N, \{\beta^c\}_{c=1}^N \in \mathbb{R}^d$, we have*

$$\|\beta - \theta - \eta(\nabla f(\beta) - \nabla f(\theta))\|_2^2 \le (1 - \eta m) \cdot \|\beta - \theta\|_2^2 + 4\eta L \sum_{c=1}^N p_c \cdot (\|\beta^c - \beta\|_2^2 + \|\theta^c - \theta\|_2^2).$$

where $\beta = \sum_{c=1}^N p_c \beta^c$, $\theta = \sum_{c=1}^N p_c \theta^c$, $\nabla f(\theta) = \sum_{c=1}^N p_c \nabla f^c(\theta^c)$, and $\nabla f(\beta) = \sum_{c=1}^N p_c \nabla f^c(\beta^c)$. We postpone the proof into Section D.1. The above result implies that as long as the local parameters $\theta^c, \beta^c$ and global $\theta, \beta$ don't differ each other too much, we can guarantee the desired convergence.

The following result ensures a bounded gap between $\bar{\theta}_s^c$ and $\bar{\theta}_{\eta\lfloor \frac{s}{\eta} \rfloor}^c$ in $\ell_2$ norm for any $s \ge 0$ and $c \in [N]$. We postpone the proof of Lemma B.2 into Section D.2.

**Lemma B.2** (Discretization error). *Assume assumptions A.1, A.2, and A.3 hold. For any $s \ge 0$, any learning rate $\eta \in (0, 2/m)$ and $\|\theta_0^c - \theta_*\|_2^2 \le d\mathcal{D}^2$ for any $c \in [N]$, the iterates of $(\bar{\theta}_s)$ based on the continuous dynamics of Eq.(19) satisfy the following estimate*

$$\mathbb{E}\|\bar{\theta}_s^c - \bar{\theta}_{\eta\lfloor \frac{s}{\eta} \rfloor}^c\|_2^2 \le 8\eta^2 d\kappa \left( \frac{\kappa\gamma^2}{d} + L\tau \right) + 16\eta d\tau.$$

The following result shows that given a finite number of local steps $K$, the divergence between $\theta^c$ in local client and $\theta$ in the center is bounded in $\ell_2$ norm. Notably, since the non-differentiable Brownian motion leads to a lower order term $O(\eta)$ instead of $O(\eta^2)$ in $\ell^2$ norm, a naïve proof may lead to a crude upper bound. We delay the proof of Lemma B.3 into Section D.3.

**Lemma B.3** (Bounded divergence, restatement of Lemma 5.5). *Assume assumptions A.1, A.2, and A.3 hold. For any learning rate $\eta \in (0, 2/m)$ and $\|\theta_0^c - \theta_*\|_2^2 \le d\mathcal{D}^2$ for any $c \in [N]$, we have the $\ell_2$ upper bound of the divergence between local clients and the center as follows*

$$\sum_{c=1}^N p_c \mathbb{E}\|\theta_k^c - \theta_k\|_2^2 \le 112(K-1)^2 \eta^2 dL^2 H_\rho + 8(K-1)\eta d\tau(\rho^2 + N(1 - \rho^2)),$$

*where $H_\rho, \kappa$ and $\gamma^2$ are defined as Definition A.4.*

The following presents a standard result for bounding the gap between $\nabla f(\theta)$ and $\nabla \widetilde{f}(\theta)$. We delay the proof of Lemma B.4 into Setion D.

**Lemma B.4** (Bounded variance). *Given assumption A.3, we have*

$$\mathbb{E}\|\nabla f(\theta) - \nabla \widetilde{f}(\theta)\|_2^2 \le d \cdot \sigma^2, \qquad \forall \theta \in \mathbb{R}^d.$$

Having all the preliminary results ready, now we present a crucial lemma for proving the convergence of all the algorithms.

**Lemma B.5** (One step update, restatement of Lemma 5.6). *Assume assumptions A.1, A.2, and A.3 hold. Consider Algorithm 4 with independently injected noise $\rho = 0$, any learning rate $\eta \in (0, \frac{1}{2L})$ and $\|\theta_0^c - \theta_*\|_2^2 \le d\mathcal{D}^2$ for any $c \in [N]$, where $\theta_*$ is the global minimum for the function $f$. Then*

$$W_2^2(\mu_{k+1}, \pi) \le \left(1 - \frac{\eta m}{2}\right) \cdot W_2^2(\mu_k, \pi) + 400\eta^2 dL^2 H_0((K-1)^2 + \kappa),$$

*where $\mu_k$ denotes the probability measure of $\theta_k$, $H_0, \kappa$ and $\gamma^2$ are defined as Definition A.4.*

*Proof.* The solution of the continuous-time process Eq.(19) follows that

$$\bar{\theta}_t = \bar{\theta}_0 - \int_0^t \nabla f(\bar{\theta}_s)\mathrm{d}s + \sqrt{2\tau} \cdot \overline{W}_t, \qquad \forall t \ge 0. \tag{20}$$

Set $t \to (k+1)\eta$ and $\bar{\theta}_0 \to \bar{\theta}_{k\eta}$ for Eq.(20) and consider a synchronous coupling such that $W_{(k+1)\eta} - W_{k\eta} := \sqrt{\eta}\xi_k$

$$\bar{\theta}_{(k+1)\eta} = \bar{\theta}_{k\eta} - \int_{k\eta}^{(k+1)\eta} \nabla f(\bar{\theta}_s)\mathrm{d}s + \sqrt{2\tau}(W_{(k+1)\eta} - W_{k\eta})$$

$$= \bar{\theta}_{k\eta} - \int_{k\eta}^{(k+1)\eta} \nabla f(\bar{\theta}_s)\mathrm{d}s + \sqrt{2\tau\eta}\xi_k. \tag{21}$$

We first denote $\zeta_k := \nabla \widetilde{f}(\theta_k) - \nabla f(\theta_k)$. Subtracting Eq.(18) from Eq.(21) yields that

$$\bar{\theta}_{(k+1)\eta} - \theta_{k+1}$$

$$= \bar{\theta}_{k\eta} - \theta_k + \eta\nabla\widetilde{f}(\theta_k) - \int_{k\eta}^{(k+1)\eta} \nabla f(\bar{\theta}_s)\mathrm{d}s$$

$$= \bar{\theta}_{k\eta} - \theta_k - \eta\left(\nabla f(\theta_k + \bar{\theta}_{k\eta} - \theta_k) - \nabla\widetilde{f}(\theta_k)\right) - \int_{k\eta}^{(k+1)\eta}\left(\nabla f(\bar{\theta}_s) - \nabla f(\bar{\theta}_{k\eta})\right)\mathrm{d}s \tag{22}$$

$$= \bar{\theta}_{k\eta} - \theta_k - \eta\Big(\underbrace{\nabla f(\theta_k + \bar{\theta}_{k\eta} - \theta_k) - \nabla f(\theta_k)}_{:=X_k}\Big) - \underbrace{\int_{k\eta}^{(k+1)\eta}\left(\nabla f(\bar{\theta}_s) - \nabla f(\bar{\theta}_{k\eta})\right)\mathrm{d}s}_{:=Y_k} + \eta\zeta_k.$$

Taking square and expectation on both sides, we have

$$\mathbb{E}\|\bar{\theta}_{(k+1)\eta} - \theta_{k+1}\|_2^2$$

$$= \mathbb{E}\|\bar{\theta}_{k\eta} - \theta_k - \eta X_k - Y_k\|_2^2 + \mathbb{E}\|\eta\zeta_k\|_2^2 + 2\eta\underbrace{\mathbb{E}\langle\bar{\theta}_{k\eta} - \theta_k - \eta X_k - Y_k, \zeta_k\rangle}_{\mathbb{E}\zeta_k = 0 \text{ and mutual independence}}$$

$$\le (1+q)\cdot\mathbb{E}\|\bar{\theta}_{k\eta} - \theta_k - \eta X_k\|_2^2 + (1+1/q)\cdot\mathbb{E}\|Y_k\|_2^2 + \mathbb{E}\|\eta\zeta_k\|_2^2$$

$$\le (1+q)\cdot\left((1-\eta m)\cdot\mathbb{E}\|\bar{\theta}_{k\eta} - \theta_k\|_2^2 + 4\eta L\sum_{c=1}^N p_c\cdot\left(\mathbb{E}\|\bar{\theta}_{k\eta}^c - \bar{\theta}_{k\eta}\|_2^2 + \mathbb{E}\|\theta_k^c - \theta_k\|_2^2\right)\right)$$

$$\quad + (1+1/q)\cdot\mathbb{E}\|Y_k\|_2^2 + \eta^2\sigma^2 d$$

$$\le (1+q)\cdot\left(\underbrace{(1-\eta m)}_{\phi}\mathbb{E}\|\bar{\theta}_{k\eta} - \theta_k\|_2^2 + 448\eta^3 d(K-1)^2 L^3 H_0 + 32(K-1)\eta^2 dL\tau N\right)$$

$$\quad + (1+1/q)\cdot\mathbb{E}\|Y_k\|_2^2 + \eta^2\sigma^2 d, \tag{23}$$

where the first inequality follows by the AM-GM inequality for any $q > 0$, the second inequality follows by Lemma B.1 and Assumption A.3. The third inequality follows by Lemma B.3 with $\rho = 0$; moreover, the continuous-time process conducts synchronization at any time step, hence $\bar{\theta}_{k\eta}^c = \bar{\theta}_{k\eta}$. Since the learning rate follows $\frac{1}{2L} \le \frac{1}{m+L} \le \frac{2}{m}$, the requirement of the learning rate for Lemma B.1 and Lemma B.3 is clearly satisfied.

Recall that $\phi = 1 - \eta m$, we get $\frac{1+\phi}{2} = 1 - \frac{1}{2}\eta m$. Choose $q = \frac{1+\phi}{2\phi} - 1$ so that $(1+q)\phi = \frac{(1+\phi)}{2} = 1 - \frac{1}{2}\eta m$. In addition, we have $1 + \frac{1}{q} = \frac{1+q}{q} = \frac{1+\phi}{1-\phi} \le \frac{2}{\eta m}$. It follows that

$$(1 + q) \cdot (1 - \eta m) \le 1 - \frac{1}{2}\eta m, \quad 1 + q \le \frac{1 - \frac{1}{2}\eta m}{1 - \eta m} \le 1.5, \quad (1 + 1/q) \le \frac{2}{m\eta}, \tag{24}$$

where the second inequality holds because $\eta \in (0, \frac{1}{2L}] \le \frac{1}{2m}$.

For the term $\mathbb{E}\|Y_k\|_2^2$ in Eq.(23), we have the following estimate

$$\begin{aligned}
\mathbb{E}\|Y_k\|_2^2 &= \mathbb{E}\left\|\int_{k\eta}^{(k+1)\eta} \left(\nabla f(\bar\theta_s) - \nabla f(\bar\theta_{k\eta})\right)\mathrm{d}s\right\|_2^2 \\
&\le \eta \int_{k\eta}^{(k+1)\eta} \mathbb{E}\left\|\nabla f(\bar\theta_s) - \nabla f(\bar\theta_{k\eta})\right\|_2^2 \mathrm{d}s \\
&= \eta \int_{k\eta}^{(k+1)\eta} \mathbb{E}\left\|\sum_{c=1}^N p_c\left(\nabla f^c(\bar\theta_s^c) - \nabla f^c(\bar\theta_{k\eta}^c)\right)\right\|_2^2 \mathrm{d}s \\
&\le \eta \int_{k\eta}^{(k+1)\eta} \sum_{c=1}^N p_c \cdot \mathbb{E}\left\|\nabla f^c(\bar\theta_s^c) - \nabla f^c(\bar\theta_{k\eta}^c)\right\|_2^2 \mathrm{d}s \\
&\le \eta L^2 \int_{k\eta}^{(k+1)\eta} \sum_{c=1}^N p_c \cdot \mathbb{E}\left\|\bar\theta_s^c - \bar\theta_{k\eta}^c\right\|_2^2 \mathrm{d}s \\
&\le \eta L^2 \int_{k\eta}^{(k+1)\eta} \left(8\eta^2 d\kappa\left(\frac{\kappa\gamma^2}{d} + L\tau\right) + 16\eta d\tau\right)\mathrm{d}s \\
&= 8\eta^4 dL^4 H_0 + 16\eta^3 L^2 d\tau, \tag{25}
\end{aligned}$$

where the first inequality follows by Hölder's inequality, the second inequality follows by Jensen's inequality, the third inequality follows by Assumption A.1, and the last inequality follows by Lemma B.2. The last equality holds since $\frac{\kappa}{d}\gamma^2 + L\tau \le LmH_0$ and $\kappa = L/m$.

Plugging Eq.(24) and Eq.(25) into Eq.(23), we have

$$\begin{aligned}
\mathbb{E}\|\bar\theta_{(k+1)\eta} - \theta_{k+1}\|_2^2 &\le (1 - \frac{\eta m}{2}) \cdot \mathbb{E}\|\bar\theta_{k\eta} - \theta_k\|_2^2 \\
&\quad + 672\eta^3 d(K-1)^2 L^3 H_0 + 48\eta^2 d(K-1)L\tau N \\
&\quad + 16\eta^3 dL^3\kappa H_0 + 32\eta^2 d\frac{L^2}{m}\tau + \eta^2\sigma^2 d.
\end{aligned}$$

Choose the specific Langevin diffusion $\bar\theta$ in stationary regime, we have $W_2^2(\mu_k, \pi) = \mathbb{E}\|\bar\theta_{k\eta} - \theta_k\|_2^2$ and $W_2^2(\mu_{k+1}, \pi) \le \mathbb{E}\|\bar\theta_{(k+1)\eta} - \theta_{k+1}\|_2^2$. Arranging the terms, we have

$$W_2^2(\mu_{k+1}, \pi) \le (1 - \frac{\eta m}{2}) \cdot W_2^2(\mu_k, \pi) + 400\eta^2 dL^2 H_0((K-1)^2 + \kappa),$$

where $\eta \le \frac{1}{2L}$, $\kappa \ge 1$, $m\tau \le L\tau \le L\tau N \le L\max_{c\in[N]} T_{c,0} \le LmH_0$, and $\sigma^2 \le L^2 H_0$ are applied to the result.

$\square$

## B.2 Convergence via independent noises

**Theorem B.6** (Restatement of Theorem 5.7). *Assume assumptions A.1, A.2, and A.3 hold. Consider Algorithm 4 with a fixed learning rate $\eta \in (0, \frac{1}{2L}]$ and $\|\theta_0^c - \theta_*\|_2^2 \le d\mathcal{D}^2$ for any $c \in [N]$, we have*

$$W_2(\mu_k, \pi) \le \left(1 - \frac{\eta m}{4}\right)^k \cdot \left(\sqrt{2d}(\mathcal{D} + \sqrt{\tau/m})\right) + 30\kappa\sqrt{\eta md} \cdot \sqrt{((K-1)^2 + \kappa)H_0}.$$

*where $\mu_k$ denotes the probability measure of $\theta_k$, $H_0$, $\kappa$ and $\gamma^2$ are defined as Definition A.4.*

*Proof.* Iteratively applying Theorem B.5 and arranging terms, we have that

$$
\begin{aligned}
W_2^2(\mu_k, \pi) &\leq \left(1 - \frac{\eta m}{2}\right)^k W_2^2(\mu_0, \pi) + \frac{1 - (1 - \frac{\eta m}{2})^k}{1 - (1 - \frac{\eta m}{2})} \left(400\eta^2 dL^2 H_0((K-1)^2 + \kappa)\right) \\
&\leq \left(1 - \frac{\eta m}{2}\right)^k W_2^2(\mu_0, \pi) + \frac{2}{\eta m} \left(400\eta^2 dL^2 H_0((K-1)^2 + \kappa)\right) \\
&\leq \left(1 - \frac{\eta m}{2}\right)^k W_2^2(\mu_0, \pi) + 800\kappa^2 \eta m d((K-1)^2 + \kappa) H_0,
\end{aligned} \tag{26}
$$

where $\kappa = \frac{L}{m}$. By Lemma F.1 and the initialization condition $\|\theta_0^c - \theta_*\|_2^2 \leq d\mathcal{D}^2$ for any $c \in [N]$, we have that

$$
W_2(\mu_0, \pi) \leq \sqrt{2d}(\mathcal{D} + \sqrt{\tau/m}).
$$

Applying the inequality $(1 - \frac{\eta m}{2}) \leq (1 - \frac{\eta m}{4})^2$ completes the proof. $\qquad\square$

**Discussions**

**Optimal choice of $K$.** To ensure the algorithm to achieve the $\epsilon$ precision based on the total number of steps $T_\epsilon$ and the learning rate $\eta$, we can set

$$
30\kappa\sqrt{\eta m d} \cdot \left(\sqrt{((K-1)^2 + \kappa)H_0}\right) \leq \frac{\epsilon}{2}
$$

$$
e^{-\frac{\eta m}{4} T_\epsilon} \cdot \left(\sqrt{2d}(\mathcal{D} + \sqrt{\tau/m})\right) \leq \frac{\epsilon}{2}.
$$

This directly leads to

$$
\eta m \leq \min\left\{\frac{m}{2L}, O\left(\frac{\epsilon^2}{d\kappa^2((K-1)^2 + \kappa)H_0}\right)\right\}, \quad T_\epsilon \geq \Omega\left(\frac{\log\left(\frac{d}{\epsilon}\right)}{m\eta}\right).
$$

Plugging into the upper bound of $\eta$, it implies that to reach the precision level $\epsilon$, it suffices to set

$$
T_\epsilon = \Omega\left(\frac{d\kappa^2((K-1)^2 + \kappa)H_0}{\epsilon^2} \cdot \log\left(\frac{d}{\epsilon}\right)\right). \tag{27}
$$

Since $H_0 = \Omega(\mathcal{D}^2 + \frac{\tau}{m})$, we observe that the number of communication rounds is around the order

$$
\frac{T_\epsilon}{K} = \Omega\left(K + \frac{\kappa}{K}\right),
$$

where the value of $\frac{T_\epsilon}{K}$ first decreases and then increases with respect to $K$, indicating that setting $K$ either too large or too small may lead to high communication costs and hurt the performance. Ideally, $K$ should be selected in the scale of $\Omega(\sqrt{\kappa})$. Combining the definition of $T_\epsilon$ in Eq.(27), this suggests an interesting result that the optimal $K$ should be in the order of $O(\sqrt{T_\epsilon})$. Similar results have been achieved by Stich (2019); Li et al. (2020c).

### B.3 Convergence via Varying Learning Rates

**Theorem B.7** (Restatement of Theorem 5.8). *Assume assumptions A.1, A.2, and A.3 hold. Consider Algorithm 4 with an initialization satisfying $\|\theta_0^c - \theta_*\|_2^2 \leq d\mathcal{D}^2$ for any $c \in [N]$ and varying learning rate following*

$$
\eta_k = \frac{1}{2L + (1/12)mk}, \qquad k = 1, 2, \cdots.
$$

*Then for any $k \geq 0$, we have*

$$
W_2(\mu_k, \pi) \leq 45\kappa\sqrt{((K-1)^2 + \kappa)H_0} \cdot (\eta_k m d)^{1/2}, \qquad \forall k \geq 0,
$$

*Proof.* We first denote
$$C_\kappa = 30\kappa\sqrt{((K-1)^2 + \kappa)H_0}.$$
Next, we proceed to show the following inequality by the induction method
$$W_2(\mu_k, \pi) \leq 1.5C_\kappa\left(\frac{d}{2L + (1/12)mk}\right)^{1/2} = 1.5C_\kappa(\eta_k md)^{1/2}, \qquad \forall k \geq 0, \qquad (28)$$
where the decreasing learning rate follows that
$$\eta_k = \frac{1}{2L + (1/12)mk}.$$
(i) For the case of $k = 0$, since
$$C_\kappa \geq 4\sqrt{\kappa}\sqrt{H_0} \geq 4\sqrt{\kappa}\sqrt{\mathcal{D}^2 + \frac{1}{m}\max_{c\in[N]} T_{c,0}} \geq 4\sqrt{\kappa/d}\left(\sqrt{d\mathcal{D}^2} + \sqrt{\frac{d}{m}\max_{c\in[N]} T_{c,0}}\right)$$
$$\geq 4\sqrt{\kappa/d}W_2(\mu_0, \pi), \qquad\qquad\qquad (29)$$
where the last inequality follows by Lemma F.1 and $\|\theta_0^c - \theta_*\|_2^2 \leq d\mathcal{D}^2$ for any $c \in [N]$.

It is clear that $W_2(\mu_0, \pi) \leq \frac{1}{4}C_\kappa\sqrt{\frac{md}{L}} \leq 1.5C_\kappa\sqrt{\eta_0 md}$ by Eq.(29).

(ii) If now that Eq.(28) holds for some $k \geq 0$, it follows by Lemma B.5 that
$$W_2^2(\mu_{k+1}, \pi) \leq \left(1 - \frac{\eta_k m}{2}\right) \cdot W_2^2(\mu_k, \pi) + 400\eta_k^2 dL^2 H_0((K-1)^2 + \kappa)$$
$$\leq \left(1 - \frac{\eta_k m}{2}\right) \cdot W_2^2(\mu_k, \pi) + \frac{\eta_k^2 m^2}{2}C_\kappa^2 d$$
$$\leq \left(1 - \frac{\eta_k m}{2}\right) \cdot 2.25C_\kappa^2 \eta_k md + \frac{\eta_k m}{3}2.25C_\kappa^2 \eta_k md$$
$$\leq \left(1 - \frac{\eta_k m}{6}\right) \cdot 2.25C_\kappa^2 \eta_k md.$$

Since $\left(1 - \frac{\eta_k m}{6}\right) \leq \left(1 - \frac{\eta_k m}{12}\right)^2$, we have
$$W_2(\mu_{k+1}, \pi) \leq \left(1 - \frac{\eta_k m}{12}\right) \cdot 1.5C_\kappa(\eta_k md)^{1/2}.$$

To prove $W_2(\mu_{k+1}, \pi) \leq 1.5C_\kappa(\eta_{k+1}md)^{1/2}$, it suffices to show $\left(1 - \frac{\eta_k m}{12}\right)\eta_k^{1/2} \leq \eta_{k+1}$, which is detailed as follows
$$\left(1 - \frac{\eta_k m}{12}\right)\eta_k^{1/2} = \frac{\sqrt{12}(24L + mk - m)}{(24L + mk)^{3/2}}$$
$$\leq \frac{\sqrt{12}(24L + mk - m)^{1/2}}{24L + mk}$$
$$\leq \frac{\sqrt{12}}{(24L + m(k+1))^{1/2}} := \eta_{k+1},$$
where the last inequality follows since
$$(24L + mk - m)(24L + mk + m)) \leq (24L + mk)^2.$$
$\qquad\qquad\qquad\qquad\qquad\qquad\qquad\qquad\qquad\qquad\qquad\qquad\qquad\qquad\qquad\qquad\qquad\qquad\qquad\square$

The above result implies that to achieve the precision $\epsilon$, we require
$$W_2(\mu_k, \pi) \leq 1.5C_\kappa\left(\frac{md}{2L + (1/12)mk}\right)^{1/2} \leq \epsilon.$$

The means that we only require $k = \Omega(\frac{d}{\epsilon^2})$ to achieve the precision $\epsilon$. By contrast, the fixed learning rate requires $T_\epsilon = \Omega\left(\frac{d}{\epsilon^2} \cdot \log(d/\epsilon)\right)$, which is much slower than the algorithm with varying learning rate by $O(\log(d/\epsilon))$ times.

### B.4 PRIVACY-ACCURACY TRADE-OFF VIA CORRELATED NOISES

Note that Algorithm 4 requires all the local clients to generate the independent noise $\xi_k^c$. Such a mechanism enjoys the convenience of the implementation and yields a potential to protect the privacy of data and alleviates the security issue. However, the scale of noises is maximized and inevitable slows down the convergence. For extensions, it can be naturally generalized to correlated noise based on a hyperparameter, namely the correlation coefficient $\rho$ between different clients. Replacing Eq.(13) with

$$\beta_{k+1}^c = \theta_k^c - \eta\nabla\widetilde{f}^c(\theta_k^c) + \sqrt{2\eta\tau\rho^2}\dot{\xi}_k + \sqrt{2\eta(1-\rho^2)\tau/p_c}\xi_k^c, \tag{30}$$

where $\dot{\xi}_k$ is a $d$-dimensional standard Gaussian vector shared by all the clients at iteration $k$, $\xi_k^c$ is a unique $d$-dimensional Gaussian vector generated by client $c \in [N]$ only. Moreover, $\dot{\xi}_k$ is dependent with $\xi_k^c$ for any $c \in [N]$. Following the same synchronization step based Eq.(14), we have

$$\theta_{k+1} = \theta_k - \eta\nabla\widetilde{f}(\theta_k) + \sqrt{2\eta\tau}\xi_k, \tag{31}$$

where $\xi_k = \rho\dot{\xi}_k + \sqrt{1-\rho^2}\sum_{c=1}^N \sqrt{p_c}\xi_k^c$. Since the variance of i.i.d variables is additive, it is clear that $\xi_k$ follows the standard $d$-dimensional Gaussian distribution. The inclusion of the correlated noise implicitly reduces the temperature and naturally yields a trade-off between federation and accuracy. We refer to the algorithm with correlated noise as the generalized Federated Averaging Langevin dynamics (gFA-LD) and present it in Algorithm 5.

Since the inclusion of correlated noise doesn't affect the formulation of Eq.(31), the algorithm property maintains the same except the scale of the temperature $\tau$ and federation are changed. Based on a target correlation coefficient $\rho \geq 0$, Eq.(30) is equivalent to applying a temperature $T_{c,\rho} = \tau(\rho^2 + (1-\rho^2)/p_c)$. In particular, setting $\rho = 0$ leads to $T_{c,0} = (1-\rho^2)/p_c$, which exactly recovers Algorithm 4; however, setting $\rho = 1$ leads to $T_{c,1} = \tau$, where the injected noise in local clients is reduced by $1/p_c$ times. Now we adjust the analysis as follows

**Theorem B.8** (Restatement of Theorem 5.9). *Assume assumptions A.1, A.2, and A.3 hold. Consider Algorithm 5 with a correlation coefficient $\rho \in [0,1]$, a fixed learning rate $\eta \in (0, \frac{1}{2L}]$ and $\|\theta_0^c - \theta_*\|_2^2 \leq d\mathcal{D}^2$ for any $c \in [N]$, we have*

$$W_2(\mu_k, \pi) \leq \left(1 - \frac{\eta m}{4}\right)^k \cdot \left(\sqrt{2d}(\mathcal{D} + \sqrt{\tau/m})\right) + 30\kappa\sqrt{\eta md} \cdot \sqrt{((K-1)^2 + \kappa)H_\rho},$$

*where $\mu_k$ denotes the probability measure of $\theta_k$, $H_\rho, \kappa$ and $\gamma^2$ are defined as Definition A.4.*

---

**Algorithm 5** Hybrid federated averaging Langevin dynamics algorithm (hFA-LD). Denote by $\theta_k^c$ the model parameter in the $c$-th client at the $k$-th step. Denote the immediate result of one step SGLD update from $\theta_k^c$ by $\beta_k^c$. $\xi_k^c$ is an independent standard $d$-dimensional Gaussian vector at iteration $k$ for each client $c \in [N]$ and $\dot{\xi}_k$ is a $d$-dimensional standard Gaussian vector shared by all the clients. $\rho$ denotes the correlation coefficient of the injected noises. A global synchronization is conducted every $K$ steps. This is a complete version of Algorithm 2.

1:
$$\beta_{k+1}^c = \theta_k^c - \eta\nabla\widetilde{f}^c(\theta_k^c) + \sqrt{2\eta\tau\rho^2}\dot{\xi}_k + \sqrt{2\eta(1-\rho^2)\tau/p_c}\xi_k^c,$$

2:
$$\theta_{k+1}^c = \begin{cases} \beta_{k+1}^c & \text{if } k+1 \bmod K \neq 0 \\ \sum_{c=1}^N p_c\beta_{k+1}^c & \text{if } k+1 \bmod K = 0. \end{cases}$$

---

## C PARTIAL DEVICE PARTICIPATION

Full device participation enjoys appealing convergence properties. However, it suffers from the straggler's effect in real-world applications, where the communication is limited by the slowest device. Partial device participation handles this issue by only allowing a small portion of devices in each communication and greatly increased the communication efficiency in a federated network.

## C.1 Unbiased sampling schemes

The first device-sampling scheme I Li et al. (2020b) selects a total of $S$ devices, where the $c$-th device is selected with a probability $p_c$. The first theoretical justification for convex optimization has been proposed by Li et al. (2020c).

**(Scheme I: with replacement).** Assume $\mathcal{S}_k = \{n_1, n_2, \cdots, n_S\}$, where $n_j \in [N]$ is a random number that takes a value of $c$ with a probability $p_c$ for any $j \in \{1, 2, \cdots, S\}$. The synchronization step follows that $\theta_k = \frac{1}{S} \sum_{c \in \mathcal{S}_k} \theta_k^c$.

Another strategy is to uniformly select $S$ devices without replacement. We follow Li et al. (2020c) and assume $S$ indices are selected uniformly without replacement and the synchronization step is the same as before. In addition, the convergence also requires an additional assumption on balanced data Li et al. (2020c).

**(Scheme II: without replacement).** Assume $\mathcal{S}_k = \{n_1, n_2, \cdots, n_S\}$, where $n_j \in [N]$ is a random number that takes a value of $c$ with a probability $\frac{1}{S}$ for any $j \in \{1, 2, \cdots, S\}$. Assume the data is balanced such that $p_1 = \cdots = p_N = \frac{1}{N}$. The synchronization step follows that $\theta_k = \frac{N}{S} \sum_{c \in \mathcal{S}_k} p_c \theta_k^c = \frac{1}{S} \sum_{c \in \mathcal{S}_k} \theta_k^c$.

---

**Algorithm 6** Hybrid federated averaging Langevin dynamics algorithm (hFA-LD) with partial device participation. $\xi_k^c$ is the independent Gaussian vector proposed by each client $c \in [N]$ and $\dot{\xi}_k$ is a unique Gaussian vector shared by all the clients. $\rho$ denotes the correlation coefficient. A global synchronization is conducted every $K$ steps. $\mathcal{S}_k$ is a subset that contains $S$ indices according to a device-sampling rule based on scheme I or II. This is a complete version of Algorithm 3.

---

1:
$$\beta_{k+1}^c = \theta_k^c - \eta \nabla \widetilde{f}^c(\theta_k^c) + \sqrt{2\eta\tau\rho^2}\dot{\xi}_k + \sqrt{2\eta(1-\rho^2)\tau/p_c}\xi_k^c,$$

2:
$$\theta_{k+1}^c = \begin{cases} \beta_{k+1}^c & \text{if } k+1 \bmod K \neq 0 \\ \sum_{c \in \mathcal{S}_{k+1}} \frac{1}{S}\beta_{k+1}^c & \text{if } k+1 \bmod K = 0. \end{cases}$$

---

**Lemma C.1** (Unbiased sampling scheme). *For any $k \bmod K = 0$ based on scheme I or II, we have*

$$\mathbb{E}\theta_k = \mathbb{E}\sum_{c \in \mathcal{S}_k} \theta_k^c = \beta_k := \sum_{c=1}^{N} p_c \beta_k^c.$$

*Proof.* According to the definition of scheme I or II, we have $\theta_k = \frac{1}{S} \sum_{c \in \mathcal{S}_k} \theta_k^c$. In what follows, $\mathbb{E}\theta_k = \frac{1}{S}\mathbb{E}\sum_{c \in \mathcal{S}_k} \theta_k^c = \frac{1}{S} \sum_{c_0 \in \mathcal{S}_k} \sum_{c=1}^{N} p_c \beta_k^c = \sum_{c=1}^{N} p_c \beta_k^c$, where $p_1 = p_2 = \cdots = p_N$ for scheme II in particular. □

## C.2 Bounded divergence based on partial device

**Lemma C.2** (Bounded divergence based on partial device). *Assume assumptions A.1, A.2, and A.3 hold. Consider Algorithm 6 with a correlation coefficient $\rho \in [0, 1]$, any learning rate $\eta \in (0, 2/m)$ and $\|\theta_0^c - \theta_*\|_2^2 \leq d\mathcal{D}^2$ for any $c \in [N]$, we have the following results*

*For Scheme I, the divergence between $\theta_k$ and $\beta_k$ is upper bounded by*

$$\mathbb{E}\|\beta_k - \theta_k\|_2^2 \leq \frac{112}{S}K^2\eta^2 dL^2 H_\rho + \frac{8}{S}K\eta d\tau(\rho^2 + N(1-\rho^2)).$$

*For Scheme II, assuming the data is balanced such that $p_1 = \cdots = p_N = \frac{1}{N}$, the divergence between $\theta_k$ and $\beta_k$ is upper bounded by*

$$\mathbb{E}\|\beta_k - \theta_k\|_2^2 \leq \frac{N-S}{S(N-1)}\left(112K^2\eta^2 dL^2 H_\rho + 8K\eta d\tau(\rho^2 + N(1-\rho^2))\right).$$

*where $H_\rho, \kappa$ and $\gamma^2$ are defined as Definition A.4.*

*Proof.* We prove the bounded divergence for the two schemes, respectively.

For **scheme I** with replacement, $\bar{\theta}_k = \sum_{c \in \mathcal{S}_k} \frac{1}{S} \beta_k^c$ for a subset of indices $\mathcal{S}_k$. Taking expectation with respect to $\mathcal{S}_k$, we have

$$\mathbb{E}\|\theta_k - \beta_k\|_2^2 = \frac{1}{S^2} \sum_{i=1}^S \mathbb{E}\|\beta_k^{n_i} - \beta_k\|_2^2 = \frac{1}{S} \sum_{c=1}^N p_c \|\beta_k^c - \beta_k\|_2^2, \tag{32}$$

where the first equality follows by the independence and unbiasedness of $\theta_k^{n_i}$ for any $i \in [S]$.

To further upper bound Eq.(32), we follow the same technique as in Lemma B.3. Since $k \bmod K = 0$, $k_0 = k - K$ is also the communication time, which yields the same $\theta_{k_0}^c$ for any $c \in [N]$. in what follows,

$$\sum_{c=1}^N p_c \|\beta_k^c - \beta_k\|_2^2 = \sum_{c=1}^N p_c \|\beta_k^c - \theta_{k_0} - (\beta_k - \theta_{k_0})\|_2^2$$

$$\leq \sum_{c=1}^N p_c \|\beta_k^c - \theta_{k_0}\|_2^2, \tag{33}$$

where the last inequality follows by $\beta_k = \sum_{c=1}^N p_c \beta_k^c$ and $\mathbb{E}\|x - \mathbb{E}x\|_2^2 \leq \mathbb{E}\|x\|_2^2$. Combining Eq.(32) and Eq.(33), we have

$$\mathbb{E}\|\theta_k - \beta_k\|_2^2 \leq \frac{1}{S} \sum_{c=1}^N p_c \|\beta_k^c - \theta_{k_0}\|_2^2$$

$$\leq \frac{1}{S} \sum_{c=1}^N p_c \|\beta_k^c - \theta_{k_0}^c\|_2^2$$

$$\leq \frac{1}{S} \sum_{c=1}^N p_c \mathbb{E} \sum_{k=k_0}^{k-1} 2K\eta^2 \left\|\nabla \widetilde{f^c}(\theta_k^c)\right\|_2^2 + 4K\eta d\tau\left(\rho^2 + (1-\rho^2)/p_c\right)$$

$$\leq \frac{1}{S} \sum_{c=1}^N p_c \left( \sum_{k=k_0}^{k-1} 2K\eta^2 \mathbb{E}\left\|\nabla \widetilde{f^c}(\theta_k^c)\right\|_2^2 + 4K\eta d\tau\left(\rho^2 + (1-\rho^2)/p_c\right) \right)$$

$$\leq \frac{28}{S} K^2 \eta^2 dL^2 H_\rho + \frac{4}{S} K\eta d\tau(\rho^2 + N(1-\rho^2))$$

where the last inequality follows a similar argument as in Lemma B.3.

For **scheme II**, given $p_1 = p_2 = \cdots = p_N = \frac{1}{N}$, we have $\theta_k = \frac{1}{S} \sum_{c \in \mathcal{S}_k} \beta_k^c$, which leads to

$$\mathbb{E}\|\theta_k - \beta_k\|_2^2 = \mathbb{E}\left\|\frac{1}{S} \sum_{c \in \mathcal{S}_k} \beta_k^c - \beta_k\right\|_2^2 = \frac{1}{S^2} \mathbb{E}\left\|\sum_{c=1}^N \mathbb{I}_{c \in \mathcal{S}_k}(\beta_k^c - \beta_k)\right\|_2^2,$$

where $\mathbb{I}_A$ is an indicator function that equals to 1 if the event $A$ happens.

Plugging the facts that $\mathbb{P}(c \in \mathcal{S}_k) = \frac{S}{N}$ and $\mathbb{P}(c_1, c_2 \in \mathcal{S}_k) = \frac{S(S-1)}{N(N-1)}$ for any $c_1 \neq c_2 \in [N]$ into the above equation, we have

$$\mathbb{E}\|\theta_k - \beta_k\|_2^2$$

$$= \frac{1}{S^2} \left[ \sum_{c \in [N]} \mathbb{P}(c \in \mathcal{S}_k) \|\beta_k^c - \beta_k\|_2^2 + \sum_{c_1 \neq c_2} \mathbb{P}(c_1, c_2 \in \mathcal{S}_k) \langle \beta_k^{c_1} - \beta_k, \beta_k^{c_2} - \beta_k \rangle \right]$$

$$= \frac{1}{SN} \sum_{c=1}^N \|\beta_k^c - \beta_k\|_2^2 + \sum_{c_1 \neq c_2} \frac{S-1}{SN(N-1)} \langle \beta_k^{c_1} - \beta_k, \beta_k^{c_2} - \beta_k \rangle$$

$$= \frac{1 - \frac{S}{N}}{S(N-1)} \sum_{c=1}^N \|\beta_k^c - \beta_k\|_2^2,$$

where the last equality holds since $\sum_{c \in [N]} \|\beta_k^c - \beta_k\|_2^2 + \sum_{c_1 \neq c_2} \langle \beta_k^{c_1} - \beta_k, \beta_k^{c_2} - \beta_k \rangle = \|\beta_k - \beta_k\|_2^2 = 0$.

Eventually, we have

$$
\begin{aligned}
\mathbb{E}\|\theta_k - \beta_k\|_2^2 &= \frac{N-S}{S(N-1)} \mathbb{E} \frac{1}{N} \sum_{c=1}^{N} \|\beta_k^c - \beta_k\|_2^2 \\
&\leq \frac{N-S}{S(N-1)} \mathbb{E} \frac{1}{N} \sum_{c=1}^{N} \|\beta_k^c - \theta_{k_0}\|_2^2 \\
&\leq \frac{N-S}{S(N-1)} \left( 28K^2 \eta^2 dL^2 H_\rho + 4K\eta d\tau (\rho^2 + N(1-\rho^2)) \right),
\end{aligned}
$$

where the first inequality follows a similar argument as in Eq.(33) and the last inequality follows by Lemma B.3.

$\square$

## C.3 Convergence via Partial Device Participation

**Theorem C.3** (Restatement of Theorem 5.10). *Assume assumptions A.1, A.2, and A.3 hold. Consider Algorithm 6 with a correlation coefficient $\rho \in [0,1]$, a fixed learning rate $\eta \in (0, \frac{1}{2L}]$ and $\|\theta_0^c - \theta_*\|_2^2 \leq d\mathcal{D}^2$ for any $c \in [N]$, we have*

$$
\begin{aligned}
W_2(\mu_k, \pi) \leq &\left(1 - \frac{\eta m}{4}\right)^k \cdot \left( \sqrt{2d}(\mathcal{D} + \sqrt{\tau/m}) \right) \\
&+ 30\kappa\sqrt{\eta m d} \cdot \sqrt{H_\rho((K-1)^2 + \kappa)} + 2\sqrt{\frac{C_K d\tau}{Sm}(\rho^2 + N(1-\rho^2))C_S},
\end{aligned}
$$

*where $C_K = \frac{\eta m K}{1 - e^{-\frac{\eta m K}{2}}}$, $C_S = 1$ for Scheme I and $C_S = \frac{N-S}{N-1}$ for Scheme II.*

*Proof.* Note that

$$
\begin{aligned}
&\mathbb{E}\|\bar{\theta}_{(k+1)\eta} - \theta_{k+1}\|_2^2 \\
&= \mathbb{E}\|\bar{\theta}_{(k+1)\eta} - \beta_{k+1} + \beta_{k+1} - \theta_{k+1}\|_2^2 \\
&= \mathbb{E}\|\bar{\theta}_{(k+1)\eta} - \beta_{k+1}\|_2^2 + \mathbb{E}\|\beta_{k+1} - \theta_{k+1}\|_2^2 + \mathbb{E}2\langle \bar{\theta}_{(k+1)\eta} - \beta_{k+1}, \beta_{k+1} - \theta_{k+1}\rangle \\
&= \mathbb{E}\|\bar{\theta}_{(k+1)\eta} - \beta_{k+1}\|_2^2 + \mathbb{E}\|\beta_{k+1} - \theta_{k+1}\|_2^2,
\end{aligned}
$$

where the last equality follows by the unbiasedness of the device-sampling scheme in Lemma C.1.

If $k + 1 \bmod K \neq 0$, we always have $\beta_{k+1} = \theta_{k+1}$ and $\mathbb{E}\|\beta_{k+1} - \theta_{k+1}\|_2^2 = 0$. Following the same argument as in Lemma B.5, both schemes lead to the one-step iterate as follows

$$
W_2^2(\mu_{k+1}, \pi) \leq (1 - \frac{\eta m}{2}) \cdot W_2^2(\mu_k, \pi) + 400\eta^2 dL^2 H_\rho((K-1)^2 + \kappa). \tag{34}
$$

If $k + 1 \bmod K = 0$, combining Lemma C.2 and Lemma B.5, we have

$$
W_2^2(\mu_{k+1}, \pi) \leq (1 - \frac{\eta m}{2}) \cdot W_2^2(\mu_k, \pi) + 450\eta^2 dL^2 H_\rho(K^2 + \kappa) + \frac{4K d\eta\tau}{S}(\rho^2 + N(1-\rho^2))C_S, \tag{35}
$$

where $C_S = 1$ for *Scheme I* and $C_S = \frac{N-S}{N-1}$ for *Scheme II*.

Repeatedly applying Eq.(34) and Eq.(35) and arranging terms, we have that

$$
W_2^2(\mu_k, \pi) \leq \left(1 - \frac{\eta m}{2}\right)^k W_2^2(\mu_0, \pi) + \frac{2}{\eta m}\left(450\eta^2 dL^2 H_\rho(K^2 + \kappa)\right)
$$

$$+ \frac{(1 - (1 - \frac{\eta m}{2})^K)^{\lfloor k/K \rfloor}}{1 - (1 - \frac{\eta m}{2})^K} \left( \frac{4Kd\eta\tau}{S}(\rho^2 + N(1 - \rho^2))C_S \right)$$

$$\leq \left(1 - \frac{\eta m}{2}\right)^k W_2^2(\mu_0, \pi) + 900\eta m d\kappa^2 H_0((K-1)^2 + \kappa)$$

$$+ \underbrace{\frac{\eta m K}{1 - e^{-\frac{\eta m K}{2}}}}_{C_K} \frac{4Kd\eta\tau}{\eta m K S}(\rho^2 + N(1 - \rho^2))C_S,$$

$$= \left(1 - \frac{\eta m}{2}\right)^k W_2^2(\mu_0, \pi) + 900\eta m d\kappa^2 H_0((K-1)^2 + \kappa)$$

$$+ \frac{4C_K d\tau}{Sm}(\rho^2 + N(1 - \rho^2))C_S,$$

where the second inequality follows by $(1 - r)^K \leq e^{-rK}$ for any $r \geq 0$.

$\square$

## D  BOUNDING CONTRACTION, DISCRETIZATION, AND DIVERGENCE

### D.1  DOMINATED CONTRACTION PROPERTY

*Proof of Lemma B.1* . Given a client index $c \in [N]$, applying Theorem 2.1.12 Nesterov (2004) leads to

$$\langle y - x, \nabla f^c(y) - \nabla f^c(x) \rangle \geq \frac{mL}{L+m} \|y - x\|_2^2 + \frac{1}{L+m} \|\nabla f^c(y) - \nabla f^c(x)\|_2^2, \quad \forall x, y \in \mathbb{R}^d. \tag{36}$$

In what follows, we have

$$\|\beta - \theta - \eta(\nabla f(\beta) - \nabla f(\theta))\|_2^2$$
$$= \|\beta - \theta\|_2^2 - 2\eta \underbrace{\langle \beta - \theta, \nabla f(\beta) - \nabla f(\theta) \rangle}_{\mathcal{I}} + \eta^2 \|\nabla f(\beta) - \nabla f(\theta)\|_2^2. \tag{37}$$

For the second item $\mathcal{I}$ in the right hand side, we have

$$\mathcal{I} = \sum_{c=1}^N p_c \langle \beta - \theta, \nabla f^c(\beta^c) - \nabla f^c(\theta^c) \rangle$$

$$= \sum_{c=1}^N p_c \langle \beta - \beta^c + \beta^c - \theta^c + \theta^c - \theta, \nabla f^c(\beta^c) - \nabla f^c(\theta^c) \rangle$$

$$= -\sum_{c=1}^N p_c \left( \langle \beta^c - \beta, \nabla f^c(\beta^c) - \nabla f^c(\theta^c) \rangle + \langle \theta - \theta^c, \nabla f^c(\beta^c) - \nabla f^c(\theta^c) \rangle \right)$$

$$+ \sum_{c=1}^N p_c \langle \beta^c - \theta^c, \nabla f^c(\beta^c) - \nabla f^c(\theta^c) \rangle$$

$$\geq -\sum_{c=1}^N p_c \cdot \left( (m+L) \|\beta^c - \beta\|_2^2 + (m+L) \|\theta^c - \theta\|_2^2 + \frac{1}{2(m+L)} \|\nabla f^c(\beta^c) - \nabla f^c(\theta^c)\|_2^2 \right)$$

$$+ \sum_{c=1}^N p_c \cdot \left( \frac{mL}{L+m} \|\beta^c - \theta^c\|_2^2 + \frac{1}{L+m} \|\nabla f^c(\beta^c) - \nabla f^c(\theta^c)\|_2^2 \right)$$

$$\geq -(m+L) \sum_{c=1}^N p_c \left( \|\beta^c - \beta\|_2^2 + \|\theta^c - \theta\|_2^2 \right) + \frac{mL}{L+m} \|\beta - \theta\|_2^2$$

$$+ \frac{1}{2(L+m)} \|\nabla f(\beta) - \nabla f(\theta)\|_2^2, \tag{38}$$

where the first inequality follows by the AM-GM inequality and Eq.(36), respectively; the last inequality follows by Jensen's inequality such that

$$\sum_{c=1}^{N} p_c \|\beta^c - \theta^c\|_2^2 \geq \left\| \sum_{c=1}^{N} p_c(\beta^c - \theta^c) \right\|_2^2 = \|\beta - \theta\|_2^2$$

$$\sum_{c=1}^{N} p_c \|\nabla f^c(\beta^c) - \nabla f^c(\theta^c)\|_2^2 \geq \left\| \sum_{c=1}^{N} p_c \left( \nabla f^c(\beta^c) - \nabla f^c(\theta^c) \right) \right\|_2^2 = \|\nabla f(\beta) - \nabla f(\theta)\|_2^2.$$

Plugging Eq.(38) into Eq.(37), we have

$$\|\beta - \theta - \eta \cdot (\nabla f(\beta) - \nabla f(\theta))\|_2^2$$

$$\leq \left(1 - \frac{2\eta m L}{m+L}\right) \cdot \|\beta - \theta\|_2^2 + \eta \underbrace{\left(\eta - \frac{1}{m+L}\right)}_{\leq 0 \text{ if } \eta \leq \frac{1}{m+L}} \cdot \|\nabla f(\beta) - \nabla f(\theta)\|_2^2$$

$$+ 2\eta(m+L) \sum_{c=1}^{N} p_c \cdot (\|\beta^c - \beta\|_2^2 + \|\theta^c - \theta\|_2^2)$$

$$\leq (1 - \eta m) \|\beta - \theta\|_2^2 + 4\eta L \sum_{c=1}^{N} p_c \cdot \left(\|\beta^c - \beta\|_2^2 + \|\theta^c - \theta\|_2^2\right),$$

where the last inequality follows by $\frac{2L}{m+L} \geq 1$, $m \leq L$, $1 - 2a \leq (1-a)^2$ for any $a$, and $\eta \in (0, \frac{1}{m+L}]$.

$\square$

## D.2 DISCRETIZATION ERROR

*Proof of Lemma B.2.* For any $s \in [0, \infty)$, there exists a certain $k \in \mathbb{N}^+$ such that $s \in [k\eta, (k+1)\eta)$. By the continuous dynamics of Eq. (19), we have

$$\bar{\theta}_s^c = \bar{\theta}_{\eta\lfloor\frac{s}{\eta}\rfloor}^c + (s - k\eta)\nabla f^c(\bar{\theta}_{\eta\lfloor\frac{s}{\eta}\rfloor}^c) + \sqrt{2\tau} \int_{k\eta}^{s} d\overline{W}_t,$$

which suggests that

$$\sup_{s\in[k\eta,(k+1)\eta)} \left\|\bar{\theta}_s^c - \bar{\theta}_{\eta\lfloor\frac{s}{\eta}\rfloor}^c\right\|_2 \leq (s - k\eta)\left\|\nabla f^c(\bar{\theta}_{\eta\lfloor\frac{s}{\eta}\rfloor}^c)\right\|_2 + \sup_{s\in[k\eta,(k+1)\eta)} \left\|\int_{k\eta}^{s} \sqrt{2\tau} d\overline{W}_t\right\|_2.$$

We first square the terms on both sides and take Young's inequality and expectation

$$\mathbb{E} \sup_{s\in[k\eta,(k+1)\eta)} \left\|\bar{\theta}_s^c - \bar{\theta}_{\eta\lfloor\frac{s}{\eta}\rfloor}^c\right\|_2^2 \leq 2\mathbb{E}\left\|(s - k\eta)\nabla f^c(\bar{\theta}_{\eta\lfloor\frac{s}{\eta}\rfloor}^c)\right\|_2^2$$

$$+ 2\mathbb{E} \sup_{s\in[k\eta,(k+1)\eta)} \left\|\int_{k\eta}^{s} \sqrt{2\tau} d\overline{W}_t\right\|_2^2.$$

Then, by Burkholder-Davis-Gundy inequality (50) and Itô isometry, we have

$$\mathbb{E} \sup_{s\in[k\eta,(k+1)\eta)} \left\|\bar{\theta}_s^c - \bar{\theta}_{\eta\lfloor\frac{s}{\eta}\rfloor}^c\right\|_2^2 \leq 2\mathbb{E}\left\|(s - k\eta)\nabla f^c(\bar{\theta}_{\eta\lfloor\frac{s}{\eta}\rfloor}^c)\right\|_2^2 + 8\sum_{i=1}^{d} \mathbb{E}\int_{k\eta}^{s} 2\tau dt$$

$$\leq 2\eta^2 \mathbb{E}\left\|\nabla f^c(\bar{\theta}_{\eta\lfloor\frac{s}{\eta}\rfloor}^c)\right\|_2^2 + 16\eta d\tau. \tag{39}$$

By Young's inequality and the smoothness assumption A.1, we have

$$\mathbb{E}\|\nabla f^c(\bar{\theta}_{\eta\lfloor\frac{s}{\eta}\rfloor}^c)\|_2^2 = \mathbb{E}\|\nabla f^c(\bar{\theta}_{\eta\lfloor\frac{s}{\eta}\rfloor}^c) - \nabla f^c(\theta_*) + \nabla f^c(\theta_*)\|_2^2$$

$$
\begin{aligned}
&\leq 2\mathbb{E}\|\nabla f^c(\bar{\theta}^c_{\eta\lfloor\frac{s}{\eta}\rfloor}) - \nabla f^c(\theta_*)\|^2_2 + 2\|\nabla f^c(\theta_*)\|^2_2 \\
&\leq 2L^2\mathbb{E}\|\bar{\theta}^c_{\eta\lfloor\frac{s}{\eta}\rfloor} - \theta_*\|^2_2 + 2\gamma^2 \\
&\leq 2L^2\left(\frac{1}{m}\left(\frac{\gamma^2}{m} + 2d\tau\right)\right) + 2\gamma^2 \\
&\leq 4d\kappa\left(\frac{\kappa\gamma^2}{d} + 4L\tau\right),
\end{aligned}
\tag{40}
$$

where the third inequality follows by Lemma E.2, the fourth step holds since $\kappa \geq 1$. Combining Eq. (39) and Eq. (40), we have

$$
\mathbb{E}\sup_{s\in[k\eta,(k+1)\eta)}\left\|\bar{\theta}^c_s - \bar{\theta}^c_{\eta\lfloor\frac{s}{\eta}\rfloor}\right\|^2_2 \leq 8\eta^2 d\kappa\left(\frac{\kappa\gamma^2}{d} + L\tau\right) + 16\eta d\tau.
$$

$\square$

### D.3 BOUNDED DIVERGENCE

*Proof of Lemma B.3.* For any $k \geq 0$, consider $k_0 = K\lfloor\frac{k}{K}\rfloor$ such that $k \leq k_0$ and $\theta^c_{k_0} = \theta_{k_0}$ for any $k \geq 0$. It is clear that $k - k_0 \leq K - 1$ for all $k \geq 0$. Consider the non-increasing learning rate such that $\eta_{k_0} \leq 2\eta_k$ for all $k - k_0 \leq K - 1$.

By the iterate Eq.(18), we have

$$
\begin{aligned}
&\sum_{c=1}^N p_c\mathbb{E}\|\theta^c_k - \theta_k\|^2_2 \\
&= \sum_{c=1}^N p_c\mathbb{E}\|\theta^c_k - \theta_{k_0} - (\theta_k - \theta_{k_0})\|^2_2 \\
&\leq \sum_{c=1}^N p_c\mathbb{E}\|\theta^c_k - \theta_{k_0}\|^2_2 \\
&\leq \sum_{c=1}^N p_c\mathbb{E}\sum_{k=k_0}^{k-1} 2(K-1)\eta^2_k\left\|\nabla\widetilde{f}^c(\theta^c_k)\right\|^2_2 + 4(K-1)\eta_k d\tau(\rho^2 + (1-\rho^2)/p_c) \\
&\leq \sum_{c=1}^N p_c\left(\sum_{k=k_0}^{k-1} 2(K-1)\eta^2_{k_0}\mathbb{E}\left\|\nabla\widetilde{f}^c(\theta^c_k)\right\|^2_2 + 4(K-1)\eta_{k_0}d\tau(\rho^2 + (1-\rho^2)/p_c)\right) \\
&\leq 112(K-1)^2\eta^2_k dL^2 H_\rho + 8(K-1)\eta_k d\tau(\rho^2 + N(1-\rho^2)),
\end{aligned}
$$

where the first inequality holds by $\mathbb{E}\|\theta - \mathbb{E}\theta\|^2_2 \leq \mathbb{E}\|\theta\|^2_2$ for a stochastic variable $\theta$; the second inequality follows by $(\sum_{i=1}^{K-1} a_i)^2 \leq (K-1)\sum_{i=1}^{K-1} a^2_i$; the last inequality follows by Lemma E.3 and $\eta^2_{k_0} \leq 4\eta^2_k$. $H_\rho$ is defined in Definition A.4.

$\square$

### D.4 BOUNDED VARIANCE

*Proof of Lemma B.4.* By assumption A.3, we have

$$
\begin{aligned}
\mathbb{E}\left\|\nabla f(\theta) - \nabla\widetilde{f}(\theta)\right\|^2_2 &= \mathbb{E}\left\|\sum_{c=1}^N p_c\left(\nabla f^c(\theta^c) - \nabla\widetilde{f}^c(\theta^c)\right)\right\|^2_2 \\
&= \sum_{c=1}^N p^2_c\mathbb{E}\left\|\nabla f^c(\theta^c) - \nabla\widetilde{f}^c(\theta^c)\right\|^2_2
\end{aligned}
$$

$$\le d\sigma^2 \sum_{c=1}^{N} p_c^2 \le d\sigma^2 \left(\sum_{c=1}^{N} p_c\right)^2 := d\sigma^2.$$

$\square$

# E  UNIFORM UPPER BOUND

## E.1  DISCRETE DYNAMICS

**Lemma E.1** (Discrete dynamics). *Assume assumptions A.1, A.2, and A.3 hold. We consider the generalized formulation in Algorithm 5 with the temperature*

$$T_{c,\rho} = \tau(\rho^2 + (1 - \rho^2)/p_c)$$

*given a correlation coefficient $\rho$. For any learning rate $\eta \in (0, 2/m)$ and $\|\theta_0^c - \theta_*\|_2^2 \le d\mathcal{D}^2$ for any $c \in [N]$, we have the $\ell_2$ norm upper bound as follows*

$$\sup_k \mathbb{E}\|\theta_k^c - \theta_*\|_2^2 \le d\mathcal{D}^2 + \frac{6d}{m}\left(\max_{c\in[N]} T_{c,\rho} + \frac{\sigma^2}{m} + \frac{\gamma^2}{md}\right),$$

*where $\gamma := \max_{c\in[N]} \|\nabla f^c(\theta_*)\|_2$ and $\theta_*$ denotes the global minimum for the function $f$.*

*Proof.* First, we consider the $k$-th iteration, where $k \in \{1, 2, \cdots, K-2, (K-1)_-\}$ and $(K-1)_-$ denotes the $(K-1)$-step before synchronization. Following the iterate of Eq.(13) in a local client of $c \in [N]$, we have

$$\mathbb{E}\|\theta_{k+1}^c - \theta_*\|_2^2$$
$$= \mathbb{E}\|\theta_k^c - \theta_* - \eta\nabla\widetilde{f}^c(\theta_k^c)\|_2^2 + \sqrt{8\eta T_{c,\rho}}\mathbb{E}\langle\theta_k^c - \theta_* - \eta\nabla\widetilde{f}^c(\theta_k^c), \xi_k\rangle + 2\eta T_{c,\rho}\mathbb{E}\|\xi_k\|_2^2$$
$$= \mathbb{E}\|\theta_k^c - \theta_* - \eta\nabla\widetilde{f}^c(\theta_k^c)\|_2^2 + 2\eta d T_{c,\rho}, \tag{41}$$

where the last equality follows from $\mathbb{E}\xi_k = 0$ and the conditional independence of $\theta_k^c - \theta_* - \widetilde{f}^c(\theta_k^c)$ and $\xi_k$. Note that

$$\mathbb{E}\|\theta_k^c - \theta_* - \eta\widetilde{f}^c(\theta_k^c)\|_2^2$$
$$= \mathbb{E}\|\theta_k^c - \theta_* - \eta\nabla f^c(\theta_k^c)\|_2^2 + \eta^2\mathbb{E}\|\nabla f^c(\theta_k^c) - \nabla\widetilde{f}^c(\theta_k^c)\|_2^2$$
$$\quad + 2\eta\mathbb{E}\langle\theta_k^c - \theta_* - \eta\nabla f^c(\theta_k^c), \nabla f^c(\theta_k^c) - \nabla\widetilde{f}^c(\theta_k^c)\rangle$$
$$= \mathbb{E}\|\theta_k^c - \theta_* - \eta\nabla f^c(\theta_k^c)\|_2^2 + \eta^2\mathbb{E}\|\nabla f^c(\theta_k^c) - \nabla\widetilde{f}^c(\theta_k^c)\|_2^2$$
$$\le \mathbb{E}\|\theta_k^c - \theta_* - \eta\nabla f^c(\theta_k^c)\|_2^2 + \eta^2 d\sigma^2, \tag{42}$$

where the first step follows from simple algebra, the second step follows from the unbiasedness of the stochastic gradient, and the last step follows from Assumption A.3. For any $q > 0$, we can upper bound the first term of Eq.(42) as follows

$$\mathbb{E}\|\theta_k^c - \theta_* - \eta\nabla f^c(\theta_k^c)\|_2^2$$
$$= \mathbb{E}\|\theta_k^c - \theta_* - \eta(\nabla f^c(\theta_k^c) - \nabla f^c(\theta_*)) - \eta\nabla f^c(\theta_*)\|_2^2$$
$$\le (1+q)\mathbb{E}\|\theta_k^c - \theta_* - \eta(\nabla f^c(\theta_k^c) - \nabla f^c(\theta_*))\|_2^2 + \eta^2\left(1 + \frac{1}{q}\right)\|\nabla f^c(\theta_*)\|_2^2$$
$$\le (1+q)\underbrace{\left(1 - \frac{\eta m}{2}\right)^2}_{\psi^2}\mathbb{E}\|\theta_k^c - \theta_*\|_2^2 + \eta^2\left(1 + \frac{1}{q}\right)\gamma^2, \tag{43}$$

where the first inequality follows by the AM-GM inequality; the second inequality is a special case of Lemma B.1 based on Assumption A.2, where no local steps is involved before the synchronization step. Similar results have been achieved in Theorem 3 Dalalyan (2017a). In addition, $\gamma := \max_{c\in[N]} \|\nabla f^c(\theta_*)\|_2$.

Choose $q = (\frac{1+\psi}{2\psi})^2 - 1$ so that $(1+q)\psi^2 = \frac{(1+\psi)^2}{4}$. Moreover, since $\psi = 1 - \frac{\eta m}{2}$, we get $\frac{1+\psi}{2} = 1 - \frac{1}{4}\eta m$. In addition, we have $1 + \frac{1}{q} = \frac{1+q}{q} = \frac{(1+\psi)^2}{(1-\psi)(1+3\psi)} \le \frac{2}{\eta m}$. It follows that

$$\eta^2 \left(1 + \frac{1}{q}\right) \le \frac{2\eta}{m}. \tag{44}$$

Combining Eq. (41), Eq. (42), Eq. (43), and Eq. (44), we have the following iterate

$$\mathbb{E}\|\theta_{k+1}^c - \theta_*\|_2^2 \le \underbrace{\left(1 - \frac{\eta m}{4}\right)^2}_{:=g(\eta)} \mathbb{E}\|\theta_k^c - \theta_*\|_2^2 + 2\eta dT_{c,\rho} + \eta^2 d\sigma^2 + \frac{2\eta\gamma^2}{m}.$$

Note that $\frac{1}{1-g(\eta)} = \frac{1}{\frac{\eta m}{2}(1-\frac{\eta m}{8})} \le \frac{3}{\eta m}$ given $\eta \in (0, \frac{2}{m})$. Recursively applying the above equation $k$ times, where $k \in \{1, 2, \cdots, K-1, K_-\}$ and $K_-$ denotes the $K$-step without synchronization, it follows that

$$\mathbb{E}\|\theta_k^c - \theta_*\|_2^2 \le g(\eta)^k \|\theta_0^c - \theta_*\|_2^2 + \frac{1 - g(\eta)^k}{1 - g(\eta)} \cdot \left(2\eta dT_{c,\rho} + \eta^2 d\sigma^2 + \frac{2\eta\gamma^2}{m}\right) \tag{45}$$

$$\le \|\theta_0^c - \theta_*\|_2^2 + \frac{3}{\eta m} \cdot \left(2\eta dT_{c,\rho} + \eta^2 d\sigma^2 + \frac{2\eta\gamma^2}{m}\right)$$

$$\le d\mathcal{D}^2 + \underbrace{\frac{6d}{m}\left(\max_{c\in[N]} T_{c,\rho} + \frac{\sigma^2}{m} + \frac{\gamma^2}{md}\right)}_{:=U},$$

where the second inequality holds by $g(\eta) \le 1$, the third inequality holds because $\|\theta_0^c - \theta_*\|_2^2 \le d\mathcal{D}^2$ for any $c \in [N]$ and $\eta < \frac{2}{m}$. In particular, the $K$-th step before synchronization yields that

$$\mathbb{E}\|\theta_{K_-}^c - \theta_*\|_2^2 \le d\mathcal{D}^2 + U. \tag{46}$$

Having all the results ready, for the $K$-local step after synchronization, applying Jensen's inequality

$$\mathbb{E}\|\theta_K^c - \theta_*\|_2^2 = \mathbb{E}\left\|\sum_{c=1}^N p_c \theta_{K_-}^c - \theta_*\right\|_2^2$$

$$\le \sum_{c=1}^N p_c \mathbb{E}\left\|\theta_{K_-}^c - \theta_*\right\|_2^2$$

$$\le d\mathcal{D}^2 + U, \tag{47}$$

Now starting from iteration $K$, we adapt the recursion of Eq.(45) for the $k$-th step, where $k \in \{K+1, \cdots, 2K-1, (2K)_-\}$ and $(2K)_-$ denotes the $2K$-step without synchronization, we have

$$\mathbb{E}\|\theta_k^c - \theta_*\|_2^2$$

$$\le g(\eta)^{k-K} \cdot \mathbb{E}\|\theta_K^c - \theta_*\|_2^2 + \frac{1 - g(\eta)^{k-K}}{1 - g(\eta)} \cdot \left(2\eta d \max_{c\in[N]} T_{c,\rho} + \eta^2 d\sigma^2 + \frac{2\eta\gamma^2}{m}\right)$$

$$\le g(\eta)^{k-K}(d\mathcal{D}^2 + U) + \frac{1 - g(\eta)^{k-K}}{m\eta/3}\frac{m\eta}{3}U$$

$$\le d\mathcal{D}^2 + g(\eta)^{k-K}U + (1 - g(\eta)^{k-K})U$$

$$\le d\mathcal{D}^2 + U, \tag{48}$$

where the second inequality follows by Eq.(47), the fact that $1 - g(\eta) \ge \eta m/3$ and $\eta \le \frac{2}{m}$, and the definition of $U$. The third one holds since $g(\eta) \le 1$.

By repeating Eq.(47) and (48), we have that for all $k \ge 0$, we can obtain the desired uniform upper bound. □

*Discussions:* Since the above result is independent of the learning rate $\eta$, it can be naturally applied to the setting with decreasing learning rates. The details are omitted.

### E.2 CONTINUOUS DIFFUSION

**Lemma E.2** (Continuous time). *Assume assumption A.2 holds. We have the $\ell_2$ norm upper bound as follows*

$$\sup_t \mathbb{E}\left\|\bar{\theta}_t^c - \theta_*\right\|_2^2 \le \frac{1}{m}\left(\frac{\gamma^2}{m} + 2d\tau\right),$$

*where $\gamma := \max_{c\in[N]} \|\nabla f^c(\theta_*)\|_2$ and $\theta_*$ denotes the global minimum for the function $f$.*

*Proof.* Since the synchronization is conducted at every time $t$, the essential temperature applied to each client is $\tau$. Let $q(\bar{\theta}_t^c) = \left\|\bar{\theta}_t^c - \theta_*\right\|_2^2$. For any time $t \ge 0$, applying Itô's lemma leads to

$$
\begin{aligned}
\mathrm{d}q(\bar{\theta}_t^c) &= -2\langle\bar{\theta}_t^c - \theta_*, \nabla f^c(\bar{\theta}_t^c)\rangle\mathrm{d}t + 2d\tau\mathrm{d}t + \sqrt{8\tau}\langle\bar{\theta}_t^c - \theta_*, \mathrm{d}\overline{W}_t\rangle \\
&= -2\langle\bar{\theta}_t^c - \theta_*, \nabla f^c(\bar{\theta}_t^c) - \nabla f^c(\theta_*) + \nabla f^c(\theta_*)\rangle\mathrm{d}t + 2d\tau\mathrm{d}t + \sqrt{8\tau}\langle\bar{\theta}_t^c - \theta_*, \mathrm{d}\overline{W}_t\rangle \\
&\le -2m\left\|\bar{\theta}_t^c - \theta_*\right\|_2^2\mathrm{d}t - 2\langle\bar{\theta}_t^c - \theta_*, \nabla f^c(\theta_*)\rangle\mathrm{d}t + 2d\tau\mathrm{d}t + \sqrt{8\tau}\langle\bar{\theta}_t^c - \theta_*, \mathrm{d}\overline{W}_t\rangle \\
&\le -2m\left\|\bar{\theta}_t^c - \theta_*\right\|_2^2\mathrm{d}t + m\left\|\bar{\theta}_t^c - \theta_*\right\|_2^2\mathrm{d}t + \frac{\|\nabla f^c(\theta_*)\|_2^2}{m}\mathrm{d}t + 2d\tau\mathrm{d}t + \sqrt{8\tau}\langle\bar{\theta}_t^c - \theta_*, \mathrm{d}\overline{W}_t\rangle \\
&\le -mq(\bar{\theta}_t^c)\mathrm{d}t + \left(\frac{\gamma^2}{m} + 2d\tau\right)\mathrm{d}t + \sqrt{8\tau}\langle\bar{\theta}_t^c - \theta_*, \mathrm{d}\overline{W}_t\rangle,
\end{aligned}
$$

where the first inequality follows by Assumption A.2; the second inequality follows by the AM-GM inequality; the third inequality follows by the definition that $\gamma^2 = \max_{c\in[N]} \|\nabla f^c(\theta_*)\|_2^2$.

In other words, we have

$$
\begin{aligned}
\mathrm{d}(e^{mt}q(\bar{\theta}_t^c)) &= me^{mt}q(\bar{\theta}_t^c)\mathrm{d}t + e^{mt}\mathrm{d}q(\bar{\theta}_t^c) \\
&\le me^{mt}q(\bar{\theta}_t^c)\mathrm{d}t + e^{mt}\left(-mq(\bar{\theta}_t^c)\mathrm{d}t + \left(\frac{\gamma^2}{m} + 2d\tau\right)\mathrm{d}t + \sqrt{8\tau}\langle\bar{\theta}_t^c - \theta_*, \mathrm{d}\overline{W}_t\rangle\right) \\
&\le e^{mt}\left(\frac{\gamma^2}{m} + 2d\tau\right)\mathrm{d}t + \sqrt{8\tau}e^{mt}\langle\bar{\theta}_t^c - \theta_*, \mathrm{d}\overline{W}_t\rangle.
\end{aligned}
$$

The solution is upper bounded by

$$e^{mt}q(\bar{\theta}_t^c) \le e^{m\cdot0}q(\bar{\theta}_0^c) + \int_0^t\left(e^{ms}\left(\frac{\gamma^2}{m} + 2d\tau\right)\mathrm{d}s + \sqrt{8\tau}e^{ms}\langle\bar{\theta}_s^c - \theta_*, \mathrm{d}\overline{W}_s\rangle\right).$$

By the martingale property of Itô integral, taking expectations yields

$$
\begin{aligned}
\mathbb{E}q(\bar{\theta}_t^c) &\le e^{-mt}\mathbb{E}q(\bar{\theta}_0^c) + e^{-mt}\left(\frac{\gamma^2}{m} + 2d\tau\right)\int_0^t e^{ms}\mathrm{d}s \\
&= e^{-mt}\mathbb{E}q(\bar{\theta}_0^c) + \frac{1 - e^{-mt}}{m}\left(\frac{\gamma^2}{m} + 2d\tau\right) \\
&\le e^{-mt}\mathbb{E}q(\bar{\theta}_0^c) + \frac{1 - e^{-mt}}{m}\left(\underbrace{\frac{\gamma^2}{m} + 2d\tau}_{:=V}\right),
\end{aligned}
\tag{49}
$$

where the last inequality follows since the synchronization is conducted at any time step $t$. Since $\bar{\theta}_0^c$ is simulated from the stationary distribution $\pi$, by Lemma 12 Durmus & Moulines (2016) or Theorem 17 Cheng et al. (2018), we have

$$\mathbb{E}q(\bar{\theta}_0^c) = \mathbb{E}\|\bar{\theta}_0^c - \theta_*\|_2^2 \le \frac{d\tau}{m} \le \frac{1}{m}\left(\frac{\gamma^2}{m} + 2d\tau\right) = \frac{V}{m},$$

which completes the proof.

$\square$

### E.3 BOUNDED GRADIENT

**Lemma E.3** (Bounded gradient in $\ell_2$ norm). *Given assumptions A.1, A.2, and A.3 hold, for any client $c$ and any learning rate $\eta \in (0, 2/m)$ and $\|\theta_0^c - \theta_*\|_2^2 \leq d\mathcal{D}^2$ for any $c \in [N]$, we have the $\ell_2$ norm upper bound as follows*

$$\mathbb{E}\|\nabla\widetilde{f}^c(\theta_k^c)\|_2^2 \leq 14dL^2 H_\rho,$$

*where $H_\rho = \mathcal{D}^2 + \frac{1}{m}\max_{c\in[N]} T_{c,\rho} + \frac{\gamma^2}{m^2 d} + \frac{\sigma^2}{m^2}$.*

*Proof.* Decompose the $\ell_2$ of the gradient as follows

$$\mathbb{E}\left\|\nabla\widetilde{f}^c(\theta_k^c)\right\|_2^2 = \mathbb{E}\left\|\nabla\widetilde{f}^c(\theta_k^c) - \nabla f^c(\theta_k^c) + \nabla f^c(\theta_k^c)\right\|_2^2$$

$$= \mathbb{E}\|\nabla f^c(\theta_k^c)\|_2^2 + \mathbb{E}\left\|\nabla\widetilde{f}^c(\theta_k^c) - \nabla f^c(\theta_k^c)\right\|_2^2$$

$$\quad + 2\mathbb{E}\left\langle\nabla\widetilde{f}^c(\theta_k^c) - \nabla f^c(\theta_k^c), \nabla f^c(\theta_k^c)\right\rangle$$

$$\leq \mathbb{E}\|\nabla f^c(\theta_k^c)\|_2^2 + \sigma^2 d$$

$$= \mathbb{E}\|\nabla f^c(\theta_k^c) - \nabla f^c(\theta_*) + \nabla f^c(\theta_*)\|_2^2 + \sigma^2 d$$

$$\leq 2\mathbb{E}\|\nabla f^c(\theta_k^c) - \nabla f^c(\theta_*)\|_2^2 + 2\mathbb{E}\left\|\nabla f^c(\theta_*)\right\|_2^2 + \sigma^2 d$$

$$\leq 2L^2\mathbb{E}\|\theta_k^c - \theta_*\|_2^2 + 2\gamma^2 + \sigma^2 d$$

$$\leq 2dL^2\mathcal{D}^2 + \frac{12dL^2}{m} \cdot \left(\max_{c\in[N]} T_{c,\rho} + \frac{\sigma^2}{m} + \frac{\gamma^2}{md}\right) + 2\gamma^2 + \sigma^2 d$$

$$\leq 14dL^2 \cdot \left(\mathcal{D}^2 + \frac{1}{m}\max_{c\in[N]} T_{c,\rho} + \frac{\gamma^2}{m^2 d} + \frac{\sigma^2}{m^2}\right) := 14dL^2 H_\rho,$$

where the first inequality follows by Assumption A.3; the second inequality follows by Young's inequality; the third inequality follows by Assumption A.1 and the definition that $\gamma := \max_{c\in[N]}\|\nabla f^c(\theta_*)\|_2$; the fourth inequality follows by Lemma E.1; the last inequality follows by $\kappa := \frac{L}{m} \geq 1$. $\square$

## F INITIAL CONDITION

**Lemma F.1** (Initial condition). *Let $\mu_0$ denote the Dirac delta distribution at $\theta_0$. Then, we have*

$$W_2(\mu_0, \pi) \leq \sqrt{2}(\|\theta_0 - \theta_*\|_2 + \sqrt{d\tau/m}).$$

*Proof.* By Cheng et al. (2018), there exists an optimal coupling between $\mu_0$ and $\pi$ such that

$$W_2^2(\mu_0, \pi) \leq \mathbb{E}_{\theta\sim\pi}[\|\theta_0 - \theta\|_2^2]$$

$$\leq 2\mathbb{E}_{\theta\sim\pi}[\|\theta_0 - \theta_*\|_2^2] + 2\mathbb{E}_{\theta\sim\pi}[\|\theta - \theta_*\|_2^2]$$

$$= 2\|\theta_0 - \theta_*\|_2^2 + 2\mathbb{E}_{\theta\sim\pi}[\|\theta - \theta_*\|_2^2]$$

$$\leq 2\|\theta_0 - \theta_*\|_2^2 + 2d\tau/m,$$

where the second step follows from triangle inequality, the last step follows from Lemma 12 Durmus & Moulines (2016) and the temperature $\tau$ is included to adapt to the time scaling. $\square$

**Burkholder-Davis-Gundy inequality** Let $\phi : [0,\infty) \to \mathbb{R}^{r\times d}$ for some positive integers $r$ and $d$. In addition, we assume $\mathbb{E}\int_0^\infty |\psi(s)|^2 \mathrm{d}s < \infty$ and let $Z(t) = \int_0^t \psi(s)\mathrm{d}W_s$, where $W_s$ is a $d$-dimensional Brownian motion. Then for all $t \geq 0$, we have

$$\mathbb{E}\sup_{0\leq s\leq t} |Z(s)|^2 \leq 4\mathbb{E}\int_0^t |\phi(s)|^2 \mathrm{d}s. \tag{50}$$

