# OpenReview forum: "On Convergence of Federated Averaging Langevin Dynamics"
_ICLR.cc/2022/Conference — ICLR 2022 Submitted_

### Official Review · Reviewer_dS2k · 2021-10-28

**Correctness:** 3
**Technical Novelty And Significance:** 2
**Empirical Novelty And Significance:** Not applicable
**Recommendation:** 3
**Confidence:** 5

**Main Review:**


Assuming strong convexity (Assumption 5.2) seems completely ridiculous in this setting. One of the major motivations for considering stochastic gradient Langevin dynamics is to improve the performance of non-convex stochastic optimization algorithms so as to escape spurious stationarity points and arrive at global extrema. In fact, this can be rigorously shown to tend towards the distribution of global extrema without convexity with non-asymptotic rate given in Raginsky et al 2017

There are no numerical experiments that demonstrate the utility of the approach. How restrictive is it to consider log-concave distributions? This should preclude most neural network training in practice, which is the primary emphasis of federated learning, as compared with more classical methodologies based upon consensus/multi-agent optimization.

Is there a practical phenomenon where standard federated learning or its variants that use Lagrange multipliers such as FedDyan and SCAFFOLD get stuck, whereas this version with randomized perturbations does not? I feel that this scenario is exactly in non-convex settings, which is excluded from consideration in this work.

It is difficult to make sense of what are the specific technical innovations in the analysis here, and how they are a departure from earlier results.

**Summary Of The Paper:**

This paper studies federated learning algorithms, where a collection of computing nodes seek to collectively optimize a global objective through parallelized updates executed at a server. Different from prior works which execute local-only stochastic gradient updates on non-convex losses, here a stochastic gradient Langevin update is developed, which additionally incorporates randomized Gaussian perturbations. Convergence theory is presented for the proposed scheme, which establishes point-wise  convergence in mean, as well as convergence in distribution according to the Wasserstein metric. Variations of the noise generating process which introduce heterogeneity and privacy preservation are also proposed and analyzed.


**Summary Of The Review:**


This work is mainly of theoretical interest, as it considers randomized perturbations in a federated averaging setting. However, its technical scope precludes the most useful instances of it in practice, and it is missing experiments. Therefore, it seems of limited usefulness in my view.

---

> ### Author Response · Authors · 2021-11-20
> **Strong convexity, novelties and empirical evaluations.**
>
> **$\textcolor{red}{\text{Q1. Convexity assumption is ridiculous}}$**
>
> Assuming (strong) convexity is a common practice in Bayesian learning, see [1,2,3].
>
> [1] Non-asymptotic convergence analysis for the Unadjusted Langevin Algorithm. Annals of Applied Probability. 2017
>
> [2] Theoretical guarantees for approximate sampling from smooth and log-concave densities. Journal of the Royal Statistical Society: Series B. 2017
>
> [3] User-friendly guarantees for the Langevin Monte Carlo with inaccurate gradient. Stochastic Processes and Their Applications. 2017
>
> **Q2. novelties and empirical evaluations**
>
> Please refer to our response to the public thread.

---

### Official Review · Reviewer_JGao · 2021-10-31

**Correctness:** 3
**Technical Novelty And Significance:** 2
**Empirical Novelty And Significance:** Not applicable
**Recommendation:** 5
**Confidence:** 3

**Main Review:**

The main contribution is the theoretical guarantees for a federated averaging Langevin algorithm for strongly log-concave distributions with non-i.i.d data.

The proof sketch is well written and clear. It seems that the proof is sound.
However, most of the analysis technique can be founded in lituerature, the proof technique is thus standard.

I think the writing has room to improve, but overall it is clear and easy to follow.

I have the following concerns:
1. In Section 3.1, I think it is better to stress the $\beta_k$ is an auxiliary sequence. Besides, how $\beta_k^c$ involves the Local Updates step in one iterate of the FedAvg algorithm seems to be incorrect. I think it should add $\theta_k c = \beta_k^c$ for $2 \le k \le K$.
2. Algorithm 1 is easy to understand since it is an approximation of FedAvg. However, Algorithm 2 is not so intuitive. The author didn’t explain why we should add an additional \textbf{shared} noise $\dot{\xi}_{k}$ and make the final noise correlated with the original noise $\xi_{k}^{c}$ in Algorithm 2. How does such a shared noise ensure privacy and what kind of definition of privacy is used ? The last paragraph of Section 4 is quite vague and amphibolous. To substantiate the point, I think more argument and discussion should be added.
3. The current presentation of main result is troublesome. The author could first introduce the main result and then provide a proof sketch. After all, not everyone is interested in the proof technique. What’s more, the analysis for algorithm 2 is deferred to Section 5.3.3, but its introduction is Section 4. I think it is somewhat too late and not helpful for readers to grasp the algorithm.
4. The first paragraph of Section 5.2 seem to have a type. $\nabla f\left(\bar{\theta}_{t}\right)$ should be defined as $\sum_{c=1}^{N} p_{c} \nabla f^{c}\left(\bar{\theta}_{t}\right)$.
5. Though this paper is theoretical, the author could still provide some numerical experiments to show the convergence of FA-LD. Those experiments can further validate the theorems, for example, the trade-off of $K$ and the effect of sampling methods. After all, FA-LD is different from FedAvg. I don’t know whether the convergence of FA-LD has been empirically studied before.

**Summary Of The Paper:**

The paper proposes a federated averaging Langevin algorithm (FA-LD) and analyzes its convergence under non-iid and partical participation settings.

**Summary Of The Review:**

The delcaimed contribution include FA-LD and its theoretical guarantees.
However, the analysis technique is commonly used in litureature, while no empirical expeirments to illustrate the effectiveness of FA-LD.
Besides, there are some presentation issues and ambiguity on motivation.
I don't think this paper is ready for publication.

---

> ### Author Response · Authors · 2021-11-21
> **Minor technical details and concerns on the experiments.**
>
> Thanks for the valuable comments.
>
> $\newline$
>
> **Q1. How $\beta_k^c$ involves the Local Updates step in one iterate of the FedAvg algorithm seems to be incorrect. k should be set to at least 2 instead of 1.$**
>
> We kindly disagree with your argument. The federated-averaging SGLD algorithm in Eq. (13)-(14) can be viewed as a standard extension of federated-averaging SGD in Eq.(9)(10) [1] with additional injected noise.
>
> $k$ can be set to 1, where the algorithm performs similarly to the standard SGLD algorithm with the highest communication cost. Such a mechanism is also smoothly reflected in Theorem B.6.
>
> We do acknowledge the importance of $\beta_k$ as an auxiliary sequence and we will stress it in the next revision.
>
> $\newline$
>
> **Q2. Why should we add shared noise to protect privacy?**
>
> We were not claiming the shared noise could protect privacy. On the contrary, as we stated in section 5.3.3, independently injected noise yields a potential to protect the privacy of data and alleviates the security issue. However, the large-scale noise **slows down the convergence**. To handle this issue, the independent noise can be generalized to correlated noise with correlation $\rho$.
>
> Comparing Theorem 5.7 with 5.9,  we observe that adopting correlated noise with correlation $\rho\in[0,1]$ could improve the upper bound caused by the inject noise from $O(\max_c\frac{\tau}{p_c})$ to $O(\tau \rho^2 + \max_c (\frac{\tau(1-\rho^2)}{p_c}))$, where $\sum_{c=1}^N p_c=1$. Briefly speaking, the upper bound can be improved **at least $N$ times** when $\rho=1$.
>
> We will emphasize this result in the next revision.
>
> [1] On the convergence of FedAvg on non-IID data. ICLR'20.
>
> $\newline$
>
> **Q3. Empirical evaluations**
>
> Please refer to our response to the public thread on **Empirical evaluations**.
>
> Hope that we have addressed your concerns. We would like to respond to any further confusion that you may have to facilitate the understanding.

---

### Official Review · Reviewer_sxs5 · 2021-11-01

**Correctness:** 3
**Technical Novelty And Significance:** 2
**Empirical Novelty And Significance:** Not applicable
**Recommendation:** 3
**Confidence:** 4

**Main Review:**

The paper has shown extensive theoretical analysis for the proposed FA-LD method, such as convergence for independent noise and varying learning rates, with or without full device participation. However, given the existing theoretical analysis for SGLD based on 2-Wasserstein distance and theoretical analysis for FedAvg with or without full device participation, the results are straightforward. Could the authors show the difficulty of the theoretical analysis in this paper?

Empirical studies are missing in the current version. The experiment section should be added to the paper to validate the theoretical findings and investigate the performance of FA-LD. Particularly, the reviewer would like to see the performance of FA-LD in terms of different local updates, learning rates, participating devices, etc, and its comparison with other distributed LD methods.

 Typo:
section 3.2, "a energy function"

**Summary Of The Paper:**

The paper proposes Federated Averaging Langevin Dynamic algorithm (FA-LD). The theoretical guarantees for the proposed algorithm are developed and their relationships with noise type, heterogeneity, and learning rate are studied.

**Summary Of The Review:**

The paper has given extensive theoretical convergence analysis for FA-LD algorithm via 2-Wasserstein. However, empirical studies to verify theoretical findings are missing in the main paper.

---

> ### Author Response · Authors · 2021-11-20
> **Novelties and empirical evaluations**
>
> Please refer to our response to the public thread.

---

### Official Review · Reviewer_jFv4 · 2021-11-02

**Correctness:** 3
**Technical Novelty And Significance:** 2
**Empirical Novelty And Significance:** Not applicable
**Recommendation:** 5
**Confidence:** 5

**Main Review:**

Strength:

1. The idea of sampling using Langevin diffusion in the federated learning setting is new.

Weakness:

 Since this is a theoretical paper I am basing my decision on its theoretical contribution.

1. Novelty in the proof of LD is incremental.  Only part of the proof which is different from the traditional proof of LD is the divergence term. But showing that this term is small (Lemma B.3) follows very simply from typical strongly convex optimization techniques. All the results corresponding to partial participation and varying rates follow almost directly  from the proof of the main theorem and does not have much theoretical novelty.

2. I liked the idea of allowing for correlated noise but I could not find the proof of theorem B.8 (theorem 5.9) in the paper. So it's difficult to gauge the difficulty of this part or even the correctness of this result. If I have missed the proof, and the authors could point me towards it, that would be great.


**Summary Of The Paper:**

In this work, the authors:

1.  propose a federated learning version of Langevin diffusion sampling.

2. develop theoretical guarantees for FA-LD for strongly log-concave distributions with non-i.i.d data and study how the injected noise and the stochastic gradient noise, the heterogeneity of data, and the varying learning rates affect the convergence.

3. analyze the partial participation setting.

**Summary Of The Review:**

The paper lacks theoretical novelty but I liked the idea of introducing LD in federated setting. I may change my decision after discussion with other reviewers.

---

> ### Author Response · Authors · 2021-11-20
> **Novelty concerns**
>
> Thanks for your valuable comments.
>
> **Q. Novelty concerns**
>
> Please refer to our response to the public thread. Our theoretical analysis relies on a crucial contraction property (Lemma B.1) with an additional divergence term depending on the number of local steps. Controlling the divergence in Lemma B.3, E.1, and E.3 is not a trivial study and requires to upper bound the $l_2$ norm in federated settings. Most importantly, such a study paves the way for conducting uncertainty estimation and making trustworthy predictions in federated settings with theoretical guarantees.
>
> Compared to SGLD, which addressed the scalability problems of GLD, FA-LD alleviates the privacy concerns with distributed clients and allows to train a joint model without sharing user data.
>
> **Q. Cannot find proof**
>
> The proof of theorem B.8 (theorem 5.9) is a direct application of theorem B.6, except that $H_0$ is extended to $H_{\rho}$ to account for the correlated noise.

---

### Official Review · Reviewer_4zTT · 2021-11-02

**Correctness:** 4
**Technical Novelty And Significance:** 3
**Empirical Novelty And Significance:** Not applicable
**Recommendation:** 6
**Confidence:** 3

**Main Review:**

The authors propose the federated averaging Langevin algorithm (FA-LD) for posterior inference. The authors present non-trivial non-asymptotic convergence analysis for FA-LD for distributions with strongly smooth and strongly convex Hamiltonian and with bounded variance of noise in the stochastic gradient. The assumption of bounded gradient in l2 norm is not required in contrast to the previous work FedAvg as well as many others in literature. The analysis of FA-LD indicates the number of global steps and the choice of the learning rate. More importantly, it shows that the number of local steps should be set roughly as the order of square root of the condition number. The authors further obtain non-trivial results in the cases of varying learning rates in each step, privacy-accuracy tradeoff via correlated Gaussian noises, and computation model with partial device participation.

The highlights of this paper are novel theoretical analysis of the FA-LD algorithm. It demonstrates how the injected noise, the data heterogeneity, and the stochastic-gradient noise affect the convergence. It would be nice if the authors could also provide some lower bound or worst case analysis for the FA-LD algorithm, to show that the current convergence guarantees are tight or close to tight. For example, how does FA-LD behave for Gaussian posteriors? Is the convergence rate in the main theorem asymptotically optimal for Gaussian?

Detailed comments:
Page 3, right before Section 4: “\tilde{f} is a unbiased estimate of f”
Shouldn’t it be “\grad \tilde{f} is an unbiased estimate of \grad f”?


**Summary Of The Paper:**

This paper studies the federated learning problem. In this model, local clients are allowed to jointly train a model without sharing user data, and the central goal is to design communication efficient algorithms. The author gives a federated averaging Langevin algorithm and provide theoretical guarantees for strongly log-concave distributions. These results provide guidance on the choice of learning rates and local updates to minimize communication cost. The authors also consider applying correlated noises and using only partial device updates, which are more applicable in practice.

**Summary Of The Review:**

Overall, I think this is a nice paper with strong theoretical results. It provides insights to the theoretical study of standard sampling algorithms in federated learning. The paper is also well-written.

---

> ### Author Response · Authors · 2021-11-23
> **Suggestions on the optimal rate**
>
> Thanks for your valuable comments.
>
> We believe this can be an interesting future direction to validate our theory and improve our paper. From another perspective, our result also matches the theory in [1]. Moreover, if we ignore the constant 2 along with the contraction constant $1-\frac{\eta m}{2}$, lemma B.5 also matches the order in Theorem 5 in [2].
>
> [1] QLSD: Quantised Langevin Stochastic Dynamics for Bayesian Federated Learning. 2021.
>
> [2] User-friendly guarantees for the Langevin Monte Carlo with inaccurate gradient. arXiv:1710.00095v3. 2018.

---

### Author Response · Authors · 2021-11-20
**Technical novelties**

We appreciate the reviewers for the valuable comments and the time investment. Since a few reviewers have raised their concerns about the novelties, we will respond here.

Gradient Langevin dynamics (GLD), the Euler-Maruyama discretization of Langevin diffusion, has been first proposed by [1] in the physics literature. It then attracted attention from computational statistics communities [2,3,4]. Despite the tremendous success achieved by GLD, it is not scalable to big data problems until the invention of stochastic gradient Langevin dynamics (SGLD) [5]. Since then, rich literature on convergence has been studied. For example, [6] studied the approximation analysis solely based on smoothness and growth conditions; [7, 8, 9] studied the convergence in (strongly) convex and non-convex settings, respectively.

The above results provide theoretical guarantees on the convergence of SGLD in a classical centralized learning regime, which, however, requires sharing user data and frequently causes privacy concerns in areas such as mobile devices and hospitals. Moreover, the **lack of uncertainty** in the current federated-averaging (FA) SGD framework often fails in trustworthy predictions and leads to safety concerns.

To resolve these issues, we proposed the FA-LD algorithm and conducted the first convergence analysis of SGLD in federated settings without sharing user data. **The key novelty is to study how the number of local steps, the injected noise and the stochastic gradient noise, the heterogeneity of data, and the varying learning rates affect the convergence of sampling algorithms when data privacy is a major concern**. We emphasize that proving the convergence of SGLD in federated learning requires extensively more effort than that of SGD and it yields the following contributions:

*1. **Given multiple local traps**, can we still ensure the bias decreases to 0 as the learning rate $\eta$ goes to 0? Note that achieving this target is not trivial as it seems, for example, HMC is known to be faster than GLD in centralized settings but only performs similarly to GLD in decentralized settings [10]. Our theoretical analysis relies on a crucial contraction property (Lemma B.1) with an additional divergence term depending on the number of local steps. We then control the divergence in Lemma B.3 to show the bias could decrease to 0 as $\eta\rightarrow 0$.*

*2. What is the optimal number of local steps to minimize the communication costs? We identify that the optimal local step follows a squared root order, which is similar to the result of [11].*

*3. Can we eliminate the assumption based on the bounded $L^2$ as in [11]? We prove the bounded $L^2$ norm in Lemma E.1 and E.3 in federated learning.*

In summary, we propose the first theoretical-guaranteed sampling algorithm for uncertainty estimation in federated learning. The convergence analysis highly depends on the data heterogeneity and the injected noises and yields concrete guidance on the selection of the optimal number of local updates.

[1] Correlation functions and computer simulations. Nuclear Physics B, 1981.

[2] R. M. Neal. Bayesian learning via stochastic dynamics. NIPS, 1993.

[3] Exponential convergence of Langevin distributions and their discrete approximations. Bernoulli, 1996.

[4] Non-asymptotic convergence analysis for the Unadjusted Langevin Algorithm. Annals of Applied Probability. 2017

[5] Bayesian learning via stochastic gradient Langevin dynamics. ICML'11.

[6] Approximation Analysis of Stochastic Gradient Langevin Dynamics by using Fokker-Planck Equation and Ito Process. ICML'14

[7] User-friendly guarantees for the Langevin Monte Carlo with inaccurate gradient. Stochastic Processes and Their Applications. 2017

[8] Non-convex learning via Stochastic Gradient Langevin Dynamics: a nonasymptotic analysis. COLT'17.

[9] Global Convergence of Langevin Dynamics Based Algorithms for Nonconvex Optimization. NeurIPS'18.

[10] Decentralized Stochastic Gradient Langevin Dynamics and Hamiltonian Monte Carlo. JMLR'2021.

[11] On the convergence of federated averaging SGD. ICLR'20.

---

> ### Author Response · Authors · 2021-11-20
> **Empirical evaluations**
>
> A large class of federated learning algorithms is based on the optimization framework for reasonable point estimates. Such a mechanism, however, fails to provide uncertainty estimations for target parameters, which casts doubt on the credibility of the predictions. To handle this issue, Federated Averaging Langevin dynamics (FA-LD) proposes to quantify the global posterior distribution with theoretical guarantees. **The empirical applications have been tested on CIFAR100 (image) and StackOverflow (text) for both mean predictions and uncertainty estimates (precision, recall, F1 score, et al.) [1], while the existing convergence guarantees are only restricted to Gaussian posteriors and a general convergence guarantee is still not clear**. To bridge this gap, we prove the convergence of FA-LD for strongly log-concave distributions on non-i.i.d data. Such a theoretical guarantee paves the way for conducting reliable credit intervals and hypothesis tests and shows a unique advantage in medical diagnoses, uncertainty estimation, and out-of-distribution detection tasks.
>
> [1] Federated Learning via Posterior Averaging: A New Perspective and Practical Algorithms. ICLR'21.

---

### Decision · Program_Chairs · 2022-01-20

**Decision:**

Reject

**Comment:**

This paper proposes a federated averaging Langevin dynamics (FA-LD) for numerical mean prediction with uncertainty quantification under the setting of federated learning. Convergence analysis for the proposed method under the smoothness and strong-convex assumptions is also provided, and the results are summarized in Theorems 5.7-5.10, each of which bounds the Wasserstein-2 distance $W_2(\mu_k,\pi)$ between the model distribution $\mu_k$ and the target distribution $\pi$ under different settings.

This paper received 5 reviews in total, with scores 6, 5, 3, 5, and 3. Some reviewers evaluated positively the novelty of the idea of using the Langevin dynamics in the federated setting, which I would also like to acknowledge. Upon reading the paper by myself, however, I find that the mathematical formulations are in some places not correct. What I think problematic is the third equation in equation (3): The right-hand side is a function of $N$ variables $\\\{\theta_k^c\\\}$, and they undergo different local updates at different clients when $k\not\equiv 0\mod K$ (i.e., the synchronization does not take place). Also $\nabla\tilde{f}^c$ is in general a nonlinear function of its argument. Therefore, the right-hand side cannot be written in general as a function of a single variable $\theta_k$ which is defined as $\theta_k=\sum_{c=1}^Np_c\theta_k^c$, making this equation incorrect. This problem would affect various parts of the arguments to follow in this paper, such as the first two equations in equation (16) on page 14, the two inline equations just after equation (16), equation (18), the second equality in the inline equation in page 15, line 1, and the third line in equation (25) on page 18, to mention a few. Thus I have to question the validity of the theoretical development in this paper.

Another point I would like to mention is that I did not understand the definition of Schemes I and II in Section 5.4. It is not stated at all that $\mathcal{S}_k$ is a random quantity here. Furthermore, the conditions "with/without replacement" are not described at all.  Still another point to mention is that I did not understand the claim in page 7, lines 30-31. Does it mean: If one knows the number $T_\epsilon$ of steps to achieve the precision $\epsilon$, then one should set the number $K$ of local steps per synchronization should be set of the order of $\sqrt{T_\epsilon}$. But $T_\epsilon$ depends on $K$, so that it would be unnatural to assume that one knows $T_\epsilon$ irrespective of $K$ in the first place.

Because of these, I would judge that this paper is not yet ready for presentation in its current form. I would therefore not be able to recommend acceptance of this paper.

Minor points:
- Citation style: The authors use throughout the paper what are called the *narrative citations* even though there are occasions where what are called the *parenthetical citations* (the author name and publication date are both enclosed in parentheses) should be used.
- page 3, line 7: is (the -> an) unbiased stochastic gradient; There are several unbiased estimators for the gradient, and what is mentioned here is only one instance of them.
- page 3, lines 23-24: The aggregation should take place not on each client but on the central server.
- page 3, line 36: a(n) energy function; a(n) unbiased estimate
- page 5, lines 17-20: The contents of Assumptions 5.1 and 5.2 are not assumptions but definitions.
- page 6, line 2: to obtain (the -> a)  lower bound
- page 6, line 18: $\mathcal{D}^2$ is undefined.
- page 8, line 39: (a -> the) probability $p_c$ if it is meant to be the one defined in page 3, line 8. Otherwise, use of the same symbol to represent different quantities should be avoided.
- page 14, line 25: mod ($E$ -> $K$) =0
- page 15, line 30: $H_\rho^2$ -> $H_\rho$